**Investigation**

# Genome-wide screen for deficiencies modifying *Cyclin G*-induced developmental instability in *Drosophila melanogaster*

Valérie Ribeiro,[1,2] Marco Da Costa,[1,2] Delphine Dardalhon-Cuménal,[1,2] Camille A. Dupont,[1,2] Jean-Michel Gibert (ID) ,[1,2] Emmanuèle Mouchel-Vielh,[1,2] Hélène Thomassin,[1,2] Neel B. Randsholt,[1,2] Vincent Debat (ID) ,[3,*] Frédérique Peronnet (ID) [1,2,*]

[1]Sorbonne Université, CNRS, Inserm, Développement Adaptation et Vieillissement, Dev2A, Paris F75005, France
[2]Sorbonne Université, CNRS, Inserm, Institut de Biologie Paris-Seine, IBPS, Paris F75005, France
[3]Institut de Systématique, Evolution, Biodiversité (ISYEB), Muséum National d'Histoire Naturelle, CNRS, Sorbonne Université, EPHE, Université des Antilles CP 50, 57 rue Cuvier, Paris 75005, France

*Corresponding authors: Vincent Debat, Institut de Systématique, Evolution, Biodiversité (ISYEB), Muséum National d'Histoire Naturelle, CNRS, Sorbonne Université, EPHE, Université des Antilles CP 50, 57 rue Cuvier, Paris 75005, France. Email: vincent.debat@mnhn.fr; Frédérique Peronnet. Sorbonne Université, CNRS, Inserm, Développement Adaptation et Vieillissement, Dev2A, Paris F75005, France. Email: frederique.peronnet@sorbonne-universite.fr

Despite long-lasting interest and research efforts, the genetic bases of developmental stability—the robustness to developmental noise—and its commonly used estimator, fluctuating asymmetry (FA), remain poorly understood. The *Drosophila melanogaster Cyclin G* gene (*CycG*) encodes a transcriptional cyclin that regulates growth and the cell cycle. Over-expression of a potentially more stable isoform of the protein (deletion of a PEST-rich domain, hereafter called CycG$^{\Delta P}$) induces extreme wing size and shape FA (i.e. high developmental noise), indicating a major disruption of developmental stability. Previous attempts to identify the genetic bases of FA have been impeded by the constitutively low level of developmental noise, limiting the power to detect any effect. Here, we leverage the extreme developmental instability induced by overexpression of *CycG$^{\Delta P}$* to explore the genetic bases of FA: we perform a genome-wide screen for deficiencies that enhance or reduce *CycG$^{\Delta P}$*-induced wing FA. 495 deficiencies uncovering 90% of the euchromatic genome were combined with a recombinant chromosome expressing *CycG$^{\Delta P}$*. We identified 13 and 16 deficiencies that respectively enhance and decrease FA. Analysis of mutants for some genes located in these deficiencies shows that Cyclin G ensures homogeneous growth of organs in synergy with the major morphogens of the wing, Dpp and Wg, as well as the Hippo and InR/TOR pathways. They also reveal that *CycG$^{\Delta P}$*-induced FA involves Larp, a potential direct interactor of Cyclin G that regulates translation at the mitochondrial membrane. This opens up new research perspectives for understanding developmental stability, suggesting a significant role for mitochondrial activity.

Keywords: developmental instability; *Drosophila*; fluctuating asymmetry (FA); Cyclin G; genetic screen

## Introduction

Developmental noise results from the inherent variability of biological processes which operates independently of genetic or environmental factors (for a review, see Debat and David 2002). Developmental stability refers to the buffering of developmental noise. In bilaterians, as both sides are influenced by the same environmental conditions and have the same genome, an estimate of developmental noise is provided by fluctuating asymmetry (FA), the slight, random deviations from perfect symmetry in paired structures. The genetic bases of developmental stability remain largely unknown. Investigation of mandible size and shape FA in a population of mice has identified a few single-locus quantitative trait loci (QTLs) (Leamy et al. 2015). In *Drosophila*, the wings, which have a highly stereotyped organization, have been commonly used to measure FA (e.g., Breuker et al. 2006; Takahashi et al. 2011b; Gomez and Norry 2012; see Debat and Peronnet 2013 for a review). Genome-wide deficiency mapping of regions responsible for developmental noise has identified deficiencies associated with increased centroid size or shape FA

(Takahashi et al. 2011a, 2011b). A few individual mutants have been reported to exhibit high FA, suggesting that they may encode genes crucial for maintaining developmental stability. These include for several members of the BMP signaling pathway (Debat et al. 2009), the proapoptotic gene *hid* (Neto-Silva et al. 2009), the *Drosophila insulin-like peptide 8* (*Dilp8*) gene and for its neuronal receptor *Lgr3* (Colombani et al. 2012, 2015; Garelli et al. 2012; Garelli 2015), genes encoding small Hsp proteins (Takahashi et al. 2010), the transcription factor Yorkie (Srivastava et al. 2020) and the *Tsp1* gene, which encodes the trehalose-6-synthase phosphate (Matsushita and Nishimura 2020). Strikingly, these genes are involved in a wide range of biological processes—including growth, signaling, stress response, and metabolism—indicating that developmental stability is unlikely to be controlled by a single master regulatory mechanism. Instead, it appears to be a multifactorial trait arising from the integration of multiple physiological pathways.

We have previously reported that the *Cyclin G* gene (*CycG*) of *Drosophila melanogaster*, which encodes a transcriptional cyclin,

is implicated in developmental noise (Debat et al. 2011; Debat and Peronnet 2013). Indeed, overexpression of $CycG^{AP}$, a modified version of Cyclin G that lacks the C-terminal PEST domain and is therefore potentially more stable, induces an unprecedented high level of size and shape FA (40-fold increase compared with control). The strongly destabilized system induced by $CycG^{AP}$ overexpression offers a unique opportunity to investigate the genetic bases of developmental stability. In this study, we used $CycG^{AP}$ overexpression to perform a genome-wide screen for deficiencies that enhance or reduce $CycG^{AP}$-induced wing size FA. 495 deficiencies uncovering about 90% of the euchromatic genome were combined with a recombinant chromosome expressing $CycG^{AP}$.

Development robustness includes both developmental stability —*i.e.*, the robustness to developmental noise—and canalization— the robustness to genetic and environmental effects (Debat and David 2002). As the relationship between these two components of robustness has long been debated (see Klingenberg 2019 for a review), we also measured the variation among individuals within genotypes as a measure of canalization, and tested its association with FA.

## Material and methods
### Genetics
Flies were raised in yeast-cornflour broth (yeast extract 75 g/L, cornflour 90 g/L, agar-agar 10.7 g/L, methyl 4-hydroxybenzoate 5.5 g/L) at 25 °C. Lines carrying deficiencies (*Df*) were from the Bloomington deficiency kit which combines deficiencies from different sources (BSC, Exelixis and DrosDel) (Cook et al. 2012; Supplementary Table 1). The BSC deficiencies numbered greater than 99 and the Exelixis deficiencies were generated in the same isogenic background (BL-6326 stock) (Hoskins et al. 2001; Cook et al. 2012; Roote and Russell 2012). The DrosDel deficiencies were generated in a different isogenic background (BL-5905 stock) (Ryder et al. 2004). These stocks were used as negative controls for the corresponding lines. The genetic backgrounds are referred to as $w^{5905}$ or $w^{6326}$. For the other 59 lines, for which the genetic background is unknown, either $w^{5905}$ or $w^{6326}$ were used interchangeably.

The mutant alleles used in this study are listed in Supplementary Table 2. We used FlyBase (release FB2025_05) to find information on phenotypes and genotypes (Arzu Öztürk–Çolak et al. 2024). The *da-Gal4,UAS-CycG^{AP}* third chromosome, obtained by genetic recombination (Debat et al. 2011), was used as a positive control. Due to high lethality and low fertility, this chromosome was maintained against the *TM6c,Sb* balancer in males at 18 °C.

To address the genetic interactions between *CycG* and each deficiency, five replicate crosses containing six deficiency or control females and five *da-Gal4, UAS-CycG^{AP}/TM6c,Sb* or *da-Gal4* males were performed. Series of experiments with up to 25 deficiencies were performed, each series including the appropriate positive and negative controls. Care was taken to keep the crosses in standardized conditions: tubes used for a given series were from the same batch, they were maintained at 25 °C on the same incubator floor, parents were transferred into new tubes each 48 h, three times, then discarded. The offspring were counted and kept in 70% ethanol. Thirty females of the desired genotype (*Df/da-Gal4, UAS-CycG^{AP}, +/da-Gal4, UAS-CycG^{AP}, Df/da-Gal4, or +/da-Gal4*) were randomly sampled (Supplementary Fig. 1).

### Measurements and statistical analysis
The wings were mounted on slides and scanned as described in Debat et al. (2011). The distance between landmarks 3 and 13 was used to estimate wing length (Supplementary Fig. 1). FA

was analyzed following the recommendations of Palmer and Strobeck (2003). As measurement error statistically behaves in a similar way as genuine FA (i.e., non directional random variation), each measurement was taken twice and FA was estimated using the FA10a index, as it allows to partition out measurement error from true FA (Palmer and Strobeck 2003). Briefly, a two-way mixed model ANOVA was applied to the data, considering "Individual" as a random effect and "side" as a fixed effect, repetitions being included in the residuals of the model. FA10a was computed as the interaction "individual × side" MS minus the residual MS, divided by the number of replicates (here two replicates). FA10a is thus analogous to a variance and as such is expressed in squared units of length. Differences in FA were then tested using standard F tests.

The nortest R package was used to test normality of the difference between left- and right-wing size (Shapiro–Wilk and Lilliefors tests) as well as skewness and kurtosis. Indeed, normality is a crucial aspect of FA distribution, as a lack thereof is often associated with phenomena interfering with FA (e.g., bimodality might indicate antisymmetry; see Palmer 1994 and Palmer and Strobeck 2003 for details). The rare outliers that deviated from normality, skewness ($< -0.5$ or $> +0.5$) or kurtosis ($< -3$ or $> +3$) were eliminated from the sample (see number of individuals in Supplementary Table 3), as they were identified as obvious errors in the measurements. When size varies among groups, differences in asymmetry might stem from positive allometric effects (i.e., larger size genotypes might present larger asymmetries). Results were thus checked for any size-dependence of FA. F-tests were performed to compare FA of the assays to the one of the series internal controls (*Df/da-Gal4, UAS-CycG^{AP}* compared with *+/da-Gal4, UAS-CycG^{AP}, Df/da-Gal4* compared with *+/da-Gal4*). Holm correction was applied within each series to account for multiple testing. Inter-individual variance of wing length represented the variance of the individual mean wing length in the same group.

## Results
### Genome-wide screen for deficiencies that modify $CycG^{AP}$-induced fluctuating asymmetry
To scan the whole genome for modification of $CycG^{AP}$-induced FA, we used a set of 499 deficiencies uncovering approximately 90% of the euchromatic genome (Supplementary Table 1). Of these, four deficiencies were too weak to be crossed (Supplementary Table 1). A primary screen was performed by crossing females of the remaining 495 deficiencies with *da-Gal4, UAS-CycG^{AP}/TM6c,Sb* males (Supplementary Fig. 1). As the FA values varied from experiment to experiment, a positive control (*w* females of the same genetic background crossed with *da-Gal4, UAS-CycG^{AP}/TM6c,Sb* males) and a negative control (*w* females of the same genetic background crossed with *da-Gal4/TM6c,Sb* males) were systematically included in each experimental series to serve as intrinsic standards. Of note, a number of deficiencies (59 out of 499) were generated in an unknown genetic background (indicated as *w* in Supplementary Fig. 1). In these cases, the results may reflect not only the hemizygous allelic effects revealed by the deficiencies but also potential influences of the genetic background. Forty-one deficiencies were lethal or sub-lethal when combined with the *da-Gal4, UAS-CycG^{AP}* chromosome, preventing FA analyses (Supplementary Table 1). Two deficiencies produced damaged wings when combined with *da-Gal4, UAS-CycG^{AP}*, rendering them unmeasurable (Supplementary Table 1). A total of 16,200 fly wings were mounted and analyzed in this

**Table 1.** Deficiencies modifying $CycG^{AP}$-induced FA.

| Chromosome arm | Deficiency name | Cytological location | Fluctuating asymmetry fold-change | Adjusted P-value |
|---|---|---|---|---|
| X | Df(1)BSC530 | 1A5;1B12 | 0.37329 | 4.68E−02 |
| | Df(1)M38-C5 | 8B;8E | 0.24024 | 2.86E−03 |
| | Df(1)BSC722 | 10B3;10E1 | 0.25570 | 6.72E−03 |
| | Df(1)FDD-0024486 | 14C4;14D1 | 0.12722 | 2.58E−06 |
| | Df(1)BSC643 | 15F9;16F1 | 0.28609 | 7.33E−03 |
| | Df(1)BSC405 | 16D5;16F6 | 0.37682 | 4.68E−02 |
| | Df(1)BSC871 | 18D7;18F2 | 0.32513 | 2.55E−02 |
| 2L | Df(2L)Exel7011 | 22E1;22F3 | 3.67005 | 4.34E−03 |
| | Df(2L)BSC354 | 26D7;26E3 | 4.32949 | 1.32E−03 |
| | Df(2L)BSC291 | 27D6;27F2 | 2.69177 | 3.96E−02 |
| | Df(2L)Exel7034 | 28E1;28F1 | 3.92670 | 2.69E−03 |
| | Df(2L)ED629 | 29B4;29E4 | 6.02764 | 5.81E−05 |
| | Df(2L)BSC277 | 34A1;34B2 | 0.39515 | 2.82E−02 |
| | Df(2L)ED1473 | 39B4;40A5 | 3.37401 | 7.77E−03 |
| | Df(2L)lt109 | h35;h35 | 2.87894 | 2.34E−02 |
| 2R | Df(2R)BSC880 | 49A9;49E1 | 3.36810 | 9.85E−04 |
| | Df(2R)ED3728 | 56D10;56E2 | 2.79286 | 2.50E−02 |
| 3L | Df(3L)ED201 | 61B1;61C1 | 9.14411 | 6.11E−07 |
| | Df(3L)ED4457 | 67E2;68A7 | 3.77715 | 3.03E−03 |
| | Df(3L)BSC12 | 69F6;70A2 | 7.46921 | 5.43E−06 |
| | Df(3L)ED4710 | 74D1;75B11 | 5.14760 | 1.95E−04 |
| | Df(3L)BSC419 | 78C2;78D8 | 0.20077 | 5.30E−04 |
| 3R | Df(3R)BSC549 | 83A6;83B6 | 0.28288 | 9.98E−03 |
| | Df(3R)ED5623 | 87E3;88A4 | 0.32086 | 1.65E−02 |
| | Df(3R)ED5705 | 88E12;89A5 | 0.33282 | 1.69E−02 |
| | Df(3R)ED6255 | 97D2;97F1 | 0.27386 | 5.53E−03 |
| | Df(3R)BSC322 | 98C3;98D7 | 0.33719 | 3.21E−02 |
| | Df(3R)BSC749 | 100B1;100C1 | 0.39540 | 9.04E−03 |
| | Df3R)ED6361 | 100C7;100E3 | 0.40559 | 1.17E−02 |

The fluctuating asymmetry fold-change corresponds to the ratio between fluctuating asymmetry of the deficiency combined to *da-Gal4, UAS-CycG$^{AP}$* and fluctuating asymmetry of *da-Gal4, UAS-CycG$^{AP}$* in the secondary screen.

primary screen. Of the 452 deficiencies whose combination with *da-Gal4, UAS-CycG$^{AP}$* produced viable flies with usable wings, most did not modify $CycG^{AP}$-induced FA (neutral deficiencies) (Supplementary Fig. 2). However, 39 and 21 deficiencies significantly decreased or increased $CycG^{AP}$-induced FA, respectively. To replicate the primary screen while measuring the FA of the deficiencies independently of $CycG^{AP}$ overexpression, we performed a secondary screen by crossing females of these 60 deficiencies with *da-Gal4, UAS-CycG$^{AP}$/TM6c,Sb* or *da-Gal4* males in order. In addition, eight neutral deficiencies were randomly selected to confirm the absence of an effect. The results of this secondary screen—a total of 4,640 fly wings were mounted and analyzed—are shown in Supplementary Table 3. Of the 68 deficiencies tested, only three, *Df(2L)BSC17*, *Df(1)BSC582*, and *Df(3L)BSC419*, increased FA when combined with *da-Gal4* in absence of $CycG^{AP}$ expression. None decreased it. Nevertheless, the increase was rather modest compared with that observed in context of $CycG^{AP}$ expression (FA fold-changes of 3.3, 3.4, and 4.8 when comparing *Df/+; +/da-Gal4* to *+/da-Gal4* of the same series, *vs* FA fold-changes of 28.5, 73.4, and 73.4 when comparing *+/UAS-CyG$^{AP}$,da-Gal4* to *+/da-Gal4* of the same series). Interestingly, *Df(3L)BSC419* partially overlaps *Df(3L)ED4978*, which has already been shown to increase wing shape FA (Takahashi et al. 2011b).

The absence of $CycG^{AP}$-induced FA modification was verified for the eight neutral deficiencies. However, the secondary screen eliminated some deficiencies whose effect on $CycG^{AP}$-induced FA was no longer significant, probably due to the variability of the internal control. Thus, the screen ended up with 29 modifier deficiencies: 16 that reproducibly decreased $CycG^{AP}$-induced FA and 13 that reproducibly enhanced it, further referred to as decreasing and enhancing deficiencies, respectively (Table 1 and Fig. 1). Surprisingly, the 13 enhancing deficiencies clustered on the

second chromosome and the left arm of the third chromosome whereas the 16 decreasing deficiencies clustered on the X chromosome and the right arm of the third chromosome (Table 1). We did not observe any significant correlation between the length of these deficiencies and the amplitude of their effect on $CycG^{AP}$-induced FA (Fig. 2a). A comparison of the secondary screen with a previous screen covering approximately 65% of the *Drosophila melanogaster* genome (Takahashi et al. 2011b) revealed that, among the 9 decreasing and 10 increasing deficiencies that are shared, none showed any alteration in wing centroid size FA, highlighting the power of the $CycG^{AP}$ genetic background to detect developmental noise (Supplementary Table 4).

The mean wing size of all *Df; da-Gal4, UAS-CycG$^{AP}$* flies was smaller than that of *Df; da-Gal4* flies (Fig. 2b and Supplementary Fig. 3). These differences in asymmetry were not due to a size-dependent effect, as wing size was smaller (and not larger) in the most asymmetric genotypes, and thus reflected genuine differences in developmental stability (Palmer and Strobeck 2003).

Lastly, we observed a positive correlation between FA and inter-individual variance mostly driven by a joint difference among broad categories of genotypes, i.e., deficiencies alone, with low levels of both FA and individual variance, and high levels in the enhancing deficiencies combined with $CycG^{A}$, the decreasing deficiencies showing intermediate values when combined to $CycG^{A}$ (Fig. 2c). As environmental conditions cannot be fully controlled and individuals are not genetically perfectly identical due to potential segregating variation in the parental lines, this inter-individual variance thus reflects minute genetic and environmental differences, and indirectly, canalization (e.g., Klingenberg 2019). This positive correlation between FA and inter-individual variance suggests that developmental stability and

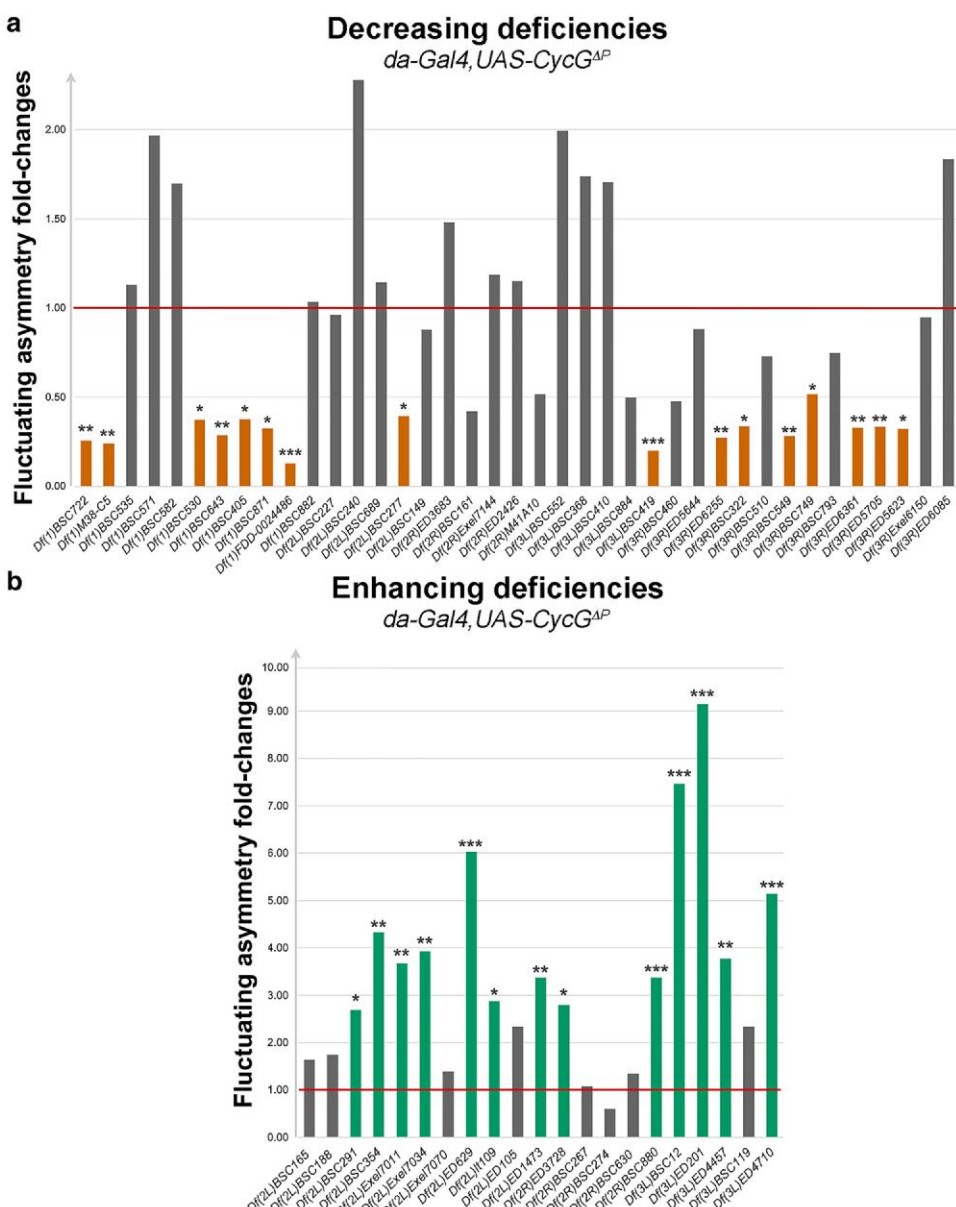

**Fig. 1.** Secondary screen—Analysis of the 60 candidate deficiencies isolated from the primary screen. a) FA of 39 deficiencies that decreased $CycG^{AP}$-induced FA in the primary screen normalised to the FA of *da-Gal4, UAS-CycG$^{AP}$* (red line). 16 of them reproduced the results of the primary screen in the secondary screen (orange). b) FA of 21 deficiencies that enhanced $CycG^{AP}$-induced FA in the primary screen normalized to the FA of *da-Gal4, UAS-CycG$^{AP}$* (red line). 13 of them reproduced the results of the primary screen in the secondary screen (green). *$P < 0.05$, **$P < 0.01$, ***$P < 0.001$.

canalization of *Drosophila* wing can be jointly affected by the combination of the deficiencies and *CycG$^{A}$*.

Taken together, these results confirmed that *CycG$^{AP}$* expression strongly perturbs developmental stability. In addition, we identified 29 deficiencies among the 452 analyzed that enhanced or decreased *CycG$^{AP}$*-induced FA and could thus contain genes important for developmental stability. These genes could therefore be partners of *CycG* in the mechanisms that maintain developmental stability.

### Identification of genes that modify CycG$^{AP}$-induced fluctuating asymmetry

We next set out to identify partners of *CycG* in developmental stability within these deficiencies. Interestingly, *Df(3R)ED6361*, which uncovers *CycG*, reduced the FA induced by *CycG$^{AP}$* expression, suggesting that expression of *CycG$^{AP}$* behaved as a dominant negative mutant and that the deviation from symmetry it induced was dose-dependent. Taken together, the enhancing deficiencies contained 736 genes (576 protein-coding genes and 160 non-protein-coding genes), and the decreasing deficiencies 865 genes (715 protein-coding genes and 150 non-protein-coding genes), making it impossible to test mutants of these genes individually. To select which mutants to test, we focused on genes encoding direct protein partners of Cyclin G or genes involved in the control of growth (Supplementary Table 2).

Although located in decreasing deficiencies (Supplementary Table 3), mutants or RNAi lines for *Aladin* (Carvalhal et al. 2015), *CG9426* (FBgn0032485), *E5* (Dalton et al. 1989), *eEF2* (Marygold et al. 2017 ), and *SAK* (Bettencourt-Dias et al. 2004) which encode direct Cyclin G physical interactors, significantly increased *CycG$^{AP}$*-induced FA (Fig. 3). Other genes, or combinations of genes, in these deficiencies may be responsible for the observed effects.

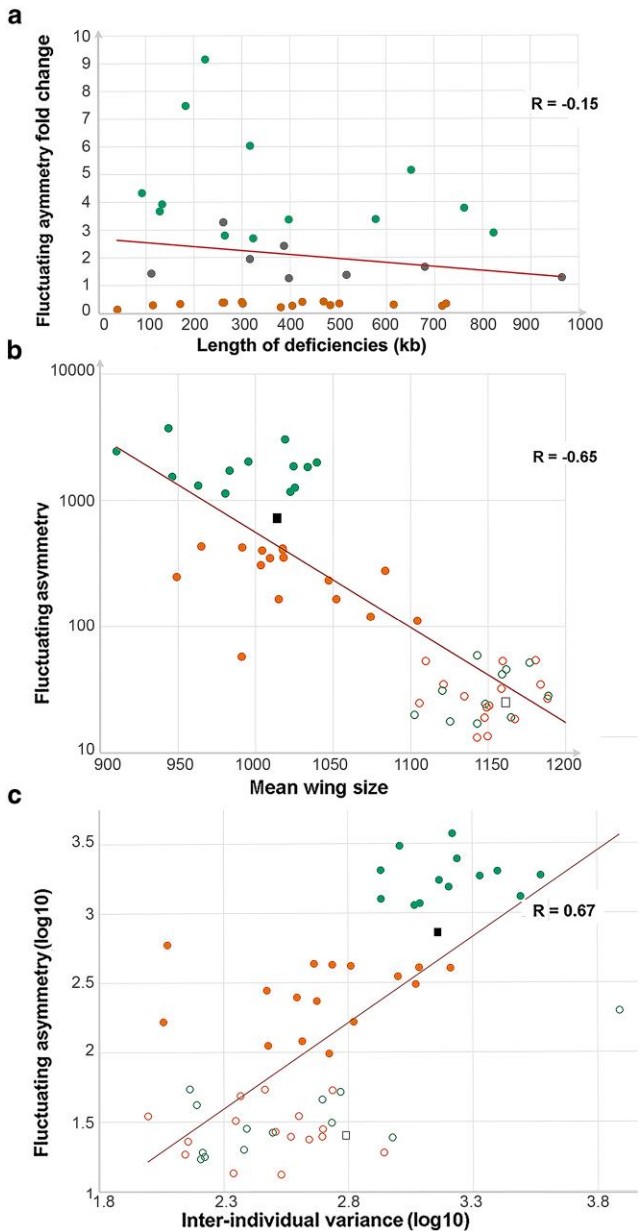

**Fig. 2.** Secondary screen—Relationship between fluctuating asymmetry and deficiency length, mean wing size and inter-individual variance. (a) FA (fold change) of the 16 decreasing (D) deficiencies (orange), the 13 enhancing (E) deficiencies (green) and 8 neutral deficiencies (gray) *vs* deficiency length. No significant correlation was observed between the two parameters (Pearson coefficient, R = −0.15). (b) FA of the 16 decreasing (D) deficiencies (orange) and the 13 enhancing (E) deficiencies (green) combined with *da-Gal4, UAS-CycG^{AP}* (filled circles) or not (empty circles) *vs* mean wing size. Filled gray square: *+/da-Gal4, UAS-CycG^{AP}*; empty grey square: *+/da-Gal4*. A strong negative correlation was observed between these parameters (R = −0.65). (c) FA (log10) of the 16 decreasing (D) deficiencies (orange) and the 13 enhancing (E) deficiencies (green) combined with *da-Gal4, UAS-CycG^{AP}* (filled circles) or not (empty circles) *vs* inter-individual variance (log10). Filled gray square: *+/da-Gal4, UAS-CycG^{AP}*; empty gray square: *+/da-Gal4*. A strong positive correlation was observed between the two parameters (R = + 0.67).

On the other hand, two RNAi lines for *larp*, which is located in a decreasing deficiency (*Df(3R)BSC322*) and also encodes a direct Cyclin G physical interactor (11), significantly reduced *CycG^{AP}*-induced FA (Fig. 3). Larp and Cyclin G may thus be direct partners in a process important for developmental stability.

The Hippo signaling pathway plays a key role in controlling organ size (10). *Merlin* (*Mer*), which encodes a positive regulator of this pathway that interacts directly with Cyclin G (11), is located in the decreasing deficiency *Df(1)BSC871*. Additionally, *hippo* (*hpo*), which encodes the kinase of the pathway, is located in *Df(2R)ED3728* that increased *CycG^{AP}*-induced FA. This prompted us to investigate the impact of other members of the Hippo pathway on *CycG^{AP}*-induced FA. Mutants for *Mer, hpo, ft, exp, wts* and *ban*, significantly increased it (Fig. 4a). Furthermore, *Mer* inactivation also increased FA independently of *CycG* deregulation (Fig. 4a). These results suggest that the Hippo pathway plays a role in developmental stability.

The *mTor* gene, which encodes the central member of another key regulator of growth (Grewal 2009), the InsulinR/TOR signaling pathway (InR/TOR), belongs to a deficiency that reduced *CycG^{AP}*-induced FA [*Df(2L)BSC277*]. Furthermore, Cyclin G interacts directly with three actors of this pathway: Akt1, PP2A and Wdb (Fischer et al. 2016; Oughtred et al. 2021). We thus tested the effect of *mTor* and other members of the pathway for their effect on *CycG^{AP}*-induced FA. Most of these, particularly mutants for *Akt1, S6K, foxo, PTEN, mTor* and *wdb*, increased *CycG^{AP}*-induced FA. This suggests that the interaction between Cyclin G and the InR/TOR pathway is crucial for developmental stability (Fig. 4b).

Finally, two major morphogens of the wing, *dpp* and *wg* [for a review see (Tabata and Takei 2004)], belonged to deficiencies that enhanced *CycG^{AP}*-induced FA [*Df(2L)Exel7011* and *Df(2L)BSC291*, respectively]. Mutants for both genes also increased *CycG^{AP}*-induced FA, indicating that we had identified at least one gene responsible for this effect within each deficiency (Fig. 4c).

## Discussion

In our efforts to identify deficiencies that modify *CycG^{AP}*-induced FA, we found that 13 deficiencies out of 452 enhanced it (enhancing deficiencies) while 16 deficiencies decreased it (decreasing deficiencies). In addition, we observed that the vast majority of modifier deficiencies (65 out of the 68 tested) do not affect FA on their own. Notably, the enhancing and decreasing deficiencies were not evenly distributed across all chromosomes. The decreasing deficiencies were enriched on chromosomes X and 3R, whereas the enhancing deficiencies were concentrated on chromosomes 2 and 3L. This pattern suggests that chromosomes 2 and 3L may have a general positive effect on developmental stability. Finally, the 13 deficiencies that enhance *CycG^{AP}*-induced FA represent only 4.1% of the *Drosophila melanogaster* genome. Collectively, these results indicate that under normal rearing conditions the control of growth is highly robust.

Overexpression of *CycG^{AP}* using the Gal4/UAS system provides an entry point to discover genes and gene networks that are essential for growth homeostasis. Unsurprisingly, our results confirm that key morphogens, Dpp and Wg, are crucial for maintaining developmental stability. A previous study had shown that insertional mutants of two BMP pathway components, *tkv*, encoding a receptor of *dpp*, and *Mad*, encoding the principal transcription factor, affect shape FA (Debat et al. 2009). Consistently, we found that both a deficiency encompassing *dpp* and *dpp* mutant alleles enhance *CycG^{AP}*-induced FA, highlighting the central role of the BMP pathway in developmental stability. We further show that the InR/Tor and Hippo growth pathways are essential for developmental stability. In line with our findings, alternative splicing of *yki*, which encodes the transcription factor Yorkie, a target of the Hippo pathway, has previously been shown to contribute to developmental stability (Srivastava et al. 2020).

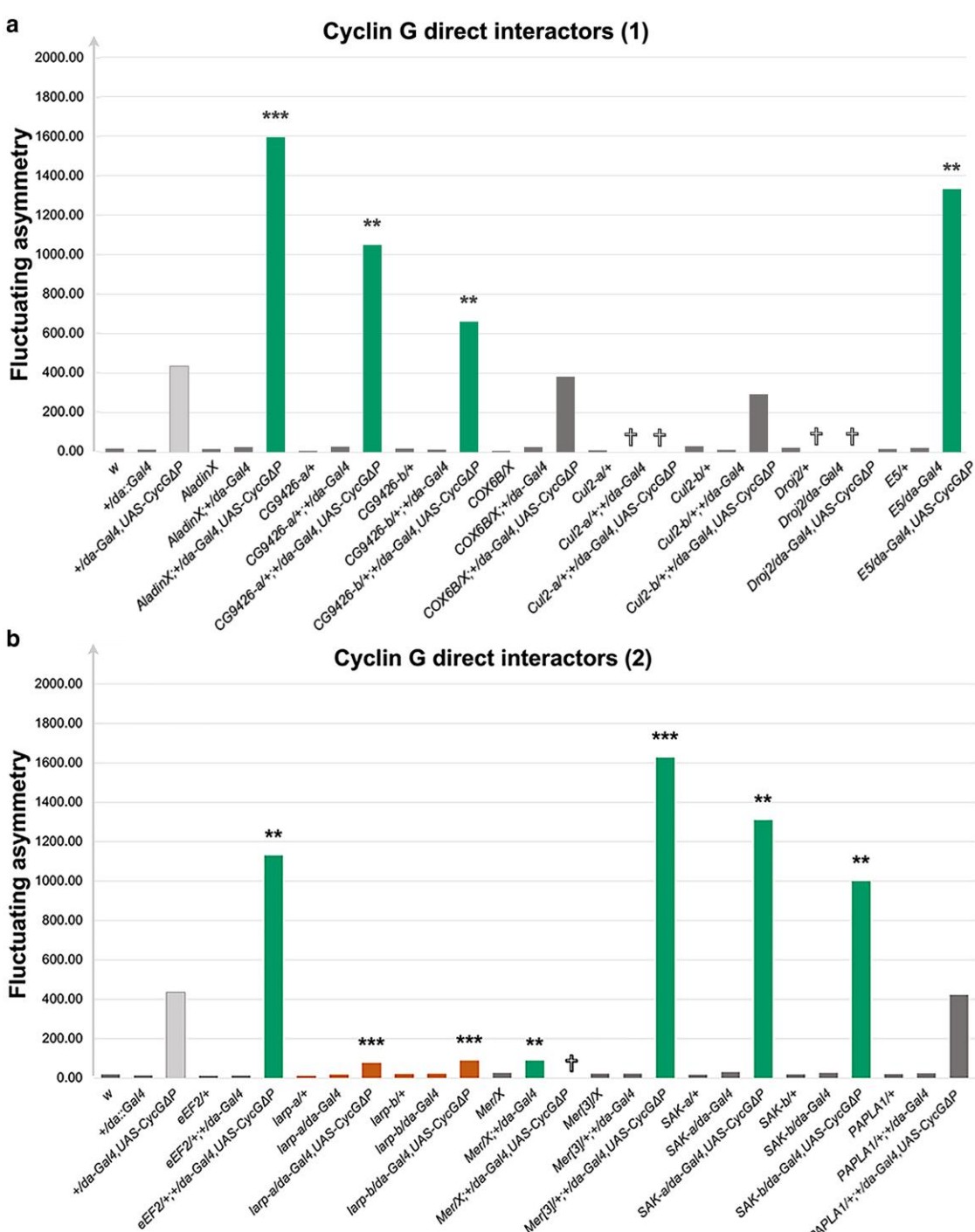

**Fig. 3.** Candidate gene analyses—genes encoding direct protein partners of Cyclin G located in deficiencies that modify *CycG^{AP}*-induced FA. In green, alleles that significantly increased *CycG^{AP}*-induced FA. In orange, alleles that significantly decreased *CycG^{AP}*-induced FA. In light gray, positive control (*+/da-Gal4, UAS-CycG^{AP}*). In dark gray, alleles that did not modify *CycG^{AP}*-induced FA. Several alleles were lethal (†) when combined to *+/da-Gal4* or *+/da-Gal4, UAS-CycG^{AP}*. *$P < 0.05$, **$P < 0.01$, ***$P < 0.001$.

Furthermore, Merlin, which promotes assembly of a functional Hippo signaling complex at the apical cell cortex, has been shown to directly interact with Cyclin G (Oughtred et al. 2021). With regard to the *Dilp8/Lgr3* axis (Colombani et al. 2012, 2015; Garelli et al. 2012; Garelli 2015), neither *Dilp8* nor *Lgr3* were found in deficiencies that modify *CycG^{AP}*-induced FA (*Df(3L)ED4685* and *Df(3L)ED4674* for *Dilp8; Df(3R)BSC321* for *Lgr3*). These results are consistent with our previous findings that expressing *CycG^{AP}* in the neuronal circuitry relaying information from the *Dilp8*

expression site to the brain does not modify FA. This led to conclude that *CycG^{AP}*-induced wing FA is an intrinsic response of the growing wing tissue (Dardalhon-Cuménal et al. 2018). Lastly, mutants of the *Tps1* gene, which is involved in trehalose metabolism, exhibit high FA that is exacerbated under dietary stress, suggesting an important role for trehalose metabolism in developmental stability (Matsushita and Nishimura 2020). *Tps1* is located within the *Df(2L)M24F-B* deficiency, which does not affect *CycG^{AP}*-induced developmental stability. Nevertheless, a more

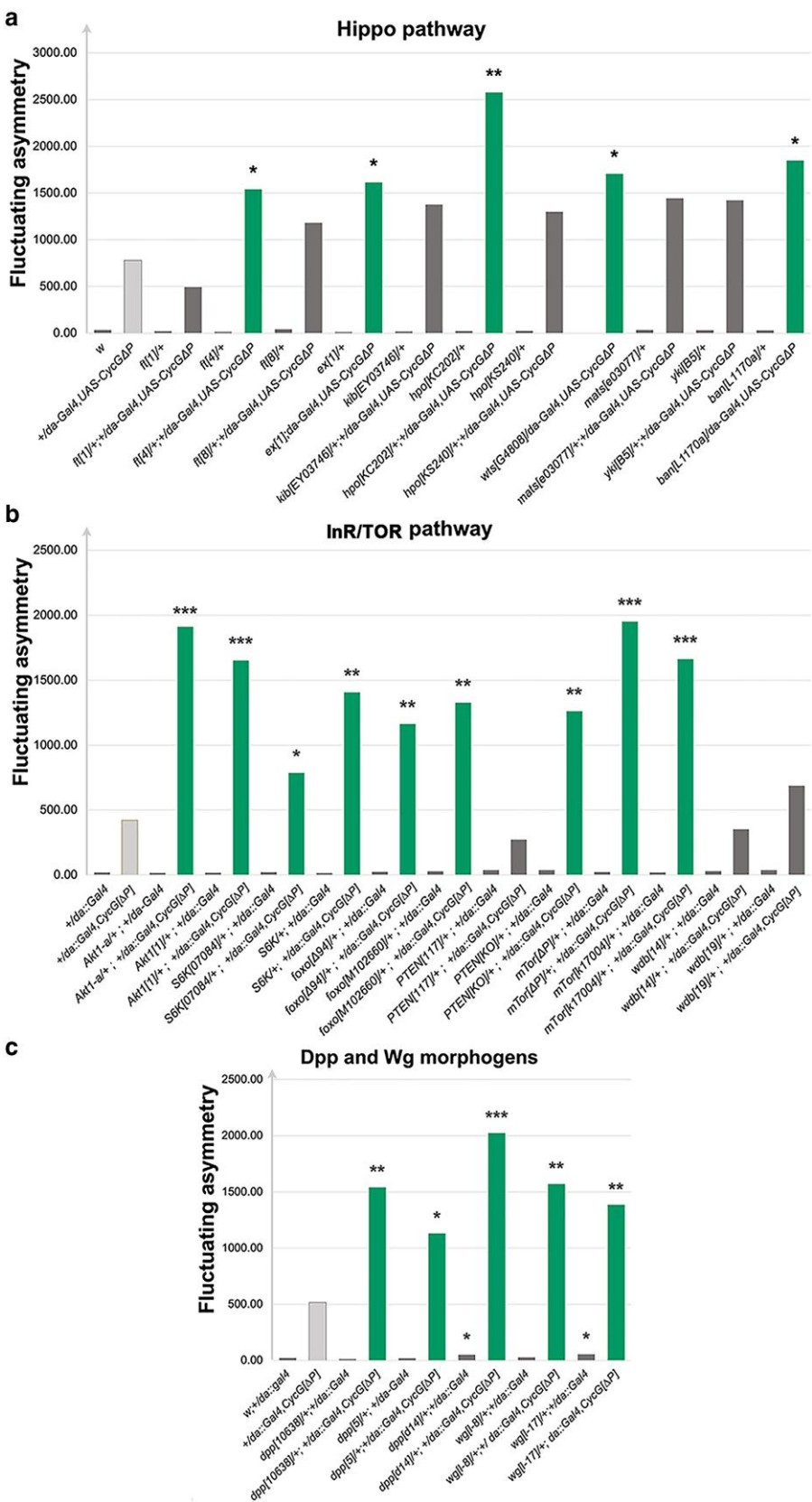

**Fig. 4.** Candidate gene analyses—Genes encoding members of the Hippo pathway (a), the InR/TOR pathway (b) and the Dpp and Wg morphogens (c). In green, alleles that significantly increased $CycG^{AP}$-induced FA. In gray, alleles that did not modify $CycG^{AP}$-induced FA. *P < 0.05, **P < 0.01, ***P < 0.001.

in-depth analysis of the interaction between *CycG* and *Tps1*, particularly under dietary stress conditions, remains to be carried out.

Beyond its specific role, Cyclin G may be unique in that it sits at the crossroads of many signaling pathways. Our previous systems biology approach has indeed shown that *CycG* is linked to many

genes in the wing imaginal disc (Dardalhon-Cuménal et al. 2018). These new findings confirm that *CycG* may be a hub within a complex genetic network important for the robustness of organ growth.

We have previously shown that expression of *CycG^{AP}* abolishes the strong negative correlation between cell size and cell number observed in wild-type fly wings (Debat et al. 2011). Cyclin G may orchestrate growth homeostasis by ensuring the proper coordination of growth and the cell cycle, which would be critical for developmental stability. This central role may explain the spectacular effects of its deregulation on FA.

Some potential direct Cyclin G partners are included in the deficiencies that modify *CycG^{AP}*-induced FA. Strikingly, these partners belong to different gene ontology categories, ranging from the aforementioned Merlin to transcription factors (e.g., E5), translation regulators (e.g., Larp) and proteins involved in protein degradation (e.g., Cul2), reinforcing the idea that Cyclin G may be involved in diverse cellular processes. Among the genes encoding these potential direct partners, *larp* is the only one whose inactivation strongly reduces *CycG^{AP}*-induced FA, suggesting that the effect of Cyclin G on FA requires this direct interaction. *Larp* encodes an evolutionarily conserved RNA-binding protein that forms a complex with the poly(A) binding protein, a translation regulator (Blagden et al. 2009; Deragon and Bousquet-Antonelli 2015; Berman et al. 2021). In addition, Larp recognizes mRNAs with a 5′ Terminal OligoPyrimidine (5′ TOP) motif, *i.e.*, mRNAs encoding proteins essential for protein synthesis such as ribosomal proteins or translation factors (Cockman et al. 2020). More recently, Larp has been shown to play a dual role as a translational repressor and a stabilizer of 5′ TOP mRNAs (Martin et al. 2022). In particular, when phosphorylated by Pink1, Larp inhibits translation at the mitochondrial outer membrane, suggesting that it may have an effect on the production of new mitochondria (Zhang et al. 2019). Interestingly, in wing imaginal discs expressing *CycG^{AP}*, genes involved in translation are up-regulated and those involved in metabolism and mitochondrial activity are down-regulated (Dardalhon-Cuménal et al. 2018). Taken together, these data support the idea that there may be a link between developmental stability and the regulation of mitochondrial activity.

## Data availability

The authors confirm that all the data necessary to verify the conclusions of the article are included in the text, figures and tables. Strains are available upon request. Raw data are deposited on Zenodo (https://doi.org/10.5281/zenodo.17416430).

Supplemental material available at GENETICS online.

## Acknowledgments

The authors would like to thank all the members of the Heterochromatin, Cell Fate and Exposome team for their valuable advice and productive discussions. They would also like to thank Dr. Y. Bellaïche for providing the *Akt1[1]* mutant, Flybase, and the Bloomington Stock Center for mutant and deficiency strains.

## Funding

This study was funded by ongoing support from the Centre National de la Recherche Scientifique (CNRS) and Sorbonne Université (SU).

Conflicts of interest. None declared.

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

*Editor: C. Peichel*