## [Peer Review File · Genetics]

Genome-wide screen for deficiencies modifying Cyclin G-induced developmental instability in *Drosophila melanogaster*.

Valerie Ribeiro, Marco Da Costa, Delphine Dardalhon-Cuménal, Camille Dupont, Jean-Michel Gibert, Emmanuèle Mouchel-Vielh, H  l  ne Thomassin, Neel Randsholt, Vincent Debat, and Frederique Peronnet

NOTE: The reviews and decision letters are unedited and appear as submitted by the reviewers.

In extremely rare instances and as determined by a Senior Editor or the EIC, portions of a review may be redacted. If a review is signed, the reviewer has agreed to no longer remain anonymous.

The review history appears in chronological order.

Review Timeline:

Submission Date:	2025-07-31
Editorial Decision:	2025-08-30
Resubmission Received:	2025-11-11
Editorial Decision:	2025-12-10
Revision Received:	2025-12-15
Accepted:	2025-12-20

August 30, 2025

GENETICS-2025-308423

Genome-wide screen for deficiencies modifying Cyclin G-induced developmental instability in *Drosophila melanogaster*.

Dear Dr. Peronnet:

Two experts in the field have reviewed your manuscript, and I have read it as well. The manuscript addresses an interesting topic and represents a huge amount of careful work. While your manuscript is not currently acceptable for publication in GENETICS, we would welcome a substantially revised manuscript. Both reviewers have comments and concerns to be addressed in a revised manuscript. You can read their reviews at the end of this email.

Both reviewers have provided detailed suggestions for improvement, and Reviewer 1 has also provided comments on the attached PDF. In addition to responding to all detailed suggestions, you should provide a much more detailed methods section, including justification of your analyses and ensure that the underlying raw data are available. In addition, the scholarship of the manuscript needs to be improved, such that previous work is properly acknowledged, and the current work is better placed into the context of what has been done before. Again, detailed suggestions are provided by Reviewer 1. We look forward to receiving your revised manuscript. Please let the editorial office know approximately how long you expect to need for revisions.

Upon resubmission, please include:

1. A clean version of your manuscript;
2. A marked version of your manuscript in which you highlight significant revisions carried out in response to the major points raised by the editor/reviewers (track changes is acceptable if preferred);
3. A detailed response to the editor's/reviewers' feedback and to the concerns listed above. Please reference line numbers in this response to aid the editor and reviewers.

Your paper will likely be sent back out for review.

Additionally, please ensure that your resubmission is formatted for GENETICS
<https://academic.oup.com/genetics/pages/general-instructions>

Follow this link to submit the revised manuscript: Link Not Available

Sincerely,

Catherine Peichel
Associate Editor
GENETICS

Approved by:
David Greenstein
Senior Editor
GENETICS

Reviewer #1 :

Please note I have also provided detailed comments on the PDF (notes associated with each highlighted segment of text).

This study was clearly a substantial body of work. Using FA to study mechanisms modulating developmental stability is really hard, requiring very careful experiments, large sample sizes, analyses and repeated measurements of samples. Despite a few issues I will discuss in detail below, the authors of this study managed to do a good job on most of these counts, so kudos! I think in principle, that once some important revisions are made, and a few additional analysis are performed to check for some potential confounding effects (in particular for cases where the modifiers of the CycG perturbation themselves impact wing size, and how this can influence the FA of the wing), this will be a valuable contribution to the genetic analysis of developmental stability in *Drosophila*.

Having said all of this I have a few major concerns that I am highlighting for the editors consideration in evaluating this manuscript for suitability for publication in Genetics.

One major issue is both context and scholarship. As I note in my direct comments on the PDF, this is not the first study using deletion mapping to identify regions that influence the magnitude of FA (the authors briefly refer to some of those studies in the results in discussion of one of the deficiencies which influences FA in the absence of the CycG perturbation). There are also a number of other studies examining specific genes and underlying biological processes and how they influence FA (and presumably developmental stability). Not just the genes that the co-senior author Dr. Debat (which I am a co-author on with Dr. Debat, so feel free to not add that as a citation) has examined (Debat et al. 2009). But also the influence of perturbation in genes like *dllp8*, *lgr3*, *Tps1*, *Hsp67*, *Yorkie* and I am sure other ones in the literature I am not aware of. So what is fundamentally new in this study is using the CycG perturbation (the over-expression of the stable version of CycG) as the basis to perform a modifier screen with deficiency mapping. So I really think the work in this study needs to be put in the larger biological context of what is known for this (in the introduction and in discussing the results of the current study). Indeed, especially because the very odd nature of the CycG perturbation (i.e. over-expression of a very stable form CycG) it is worth putting how this perturbation increases developmental instability as compared to some of the other mechanisms (in particular the global aspects for both the *dllp8/lgr3* and the *trehalose/Tps1* side of things).

My second major issue is the incredible brevity of the material and methods. I had to back track to several other studies to figure out what was going on, and also do some sleuthing in the supplemental tables of the current manuscript. All of the specific issues are documented on the PDF. Please include a reasonable amount of detail so that readers can focus on reading study to understand what you did. As it turns out, what you did (after I did my sleuthing) in terms of experimental designs, controls (mostly), sampling, repeated measuring and choice of measure for FA are good, so make that all clear to the readers so they can see that.

My only major concern methodologically has to do with some of the deletions including in this modifier screen. Most are from either the *DrosDel* or *Exelixis* deletion collections, and it seems like the authors used the corresponding co-isogenic control lines for each deletion collection as appropriate. However, some of the deletions are deficiencies that do not belong to either collection (but are available at the stock centre nonetheless). So in these cases, it is not clear what the control strains would be. So I think these strains need to be identified and highlighted as having potential confounds of both the hemizygous allelic effects uncovered by the deletion, but also a distinct genetic background for which there is no corresponding co-isogenic control strain. It may be that these still uncover really interesting modifiers, but it also may be that these effects are epistatic with the larger genetic background effects of these orphan deletion strains. I think it is a relatively modest (10%?) of the deletion strains you used, but clarifying all of this, and providing the appropriate caveats will be very important.

Some other minor comments. I think these are all on the PDF, but just for emphasis.

Did you use FA10a or FA10b (I didn't name these things).. Also add a citation to the 2003 Palmer and Strobeck book chapter. P&S 86 is a classic, but the book chapter is much clearer for people coming into the field.

I highly suggest a density plot or histogram of the distributions (for the various controls, including the CycG perturbations without the deletions) of FAs to give us a better sense of what it looked like for the primary and secondary screen. How extreme are the ones you focused on? These plots will help the readers a great deal.

Given that Kazuo Takahashi did the deletion screens previously, why not do a more formal comparison of what modifiers you identified and what regions they uncovered in there numerous studies on this? Could be really interesting.

Depending on which version of FA10 you used (see above), you may still have to contend with the dreaded effects of trait size vs trait asymmetry magnitudes issues. Examining the relationships between your FA10 measure and how much smaller or larger the mean trait size gets for those modifiers (relative to the reduction that the CycG perturbation causes on its own). Hopefully little association, but this needs to be examined empirically. Obviously $\log L - \log R$ really helps with this issue, but does not make it all go away.

I enjoyed reading this and hopefully my review is helpful,
All the best,
Ian Dworkin

Reviewer #2 :

This manuscript reports a useful screen for deficiencies and genes that interact with a fluctuating asymmetry inducing mutation, CycG ΔP to change levels of FA. The results show that quite a few regions affect FA, and that major morphogen loci within those deficiencies potentially contribute to this. In addition, mutations or knockdowns of genes in the major developmental signaling pathways outside of the deficiencies interacting with CycG ΔP also interacted with CycG ΔP to affect FA.

Altogether the results are an interesting compilation of candidates for genes potentially responsible for variation in FA.

There are two issues raised by the results that the manuscript does not mention that I think would enhance the manuscript. The first is that the results suggest that variation in FA can arise through any of the genetic processes that affect growth - at least there is no pathway the included in the experiments that did not affect FA in this background with heightened sensitivity. This supports the idea that FA is likely to be highly polygenic. This has implications for the mechanisms underlying FA, which seem unlikely to be the result of some master regulatory process.

The second is that it remains unclear whether CycG is the only way to globally destabilize the phenotype. This is not a criticism of the experiments - they were not designed to address this question. However, it is of interest to realize that this is not necessarily unique in this way. We just do not know.

The statistical methods are not really explained. Please explain the FA10 index, and why it was chosen over alternatives. The wing lengths were each measured twice, but how were these repeated measures incorporated into the analysis? How repeatable were the lengths? If repeatability is high, it might be acceptable to simply average measurements, but not if it is low. What is a 'series', in other words what group of results was the Hom-Bonferroni corrections applied to? F ratios of what exactly were used to test for differences in FA? The outlier removal procedure is ad hoc, depending on effects on summary statistics about the distribution assuming that normality is an appropriate null hypothesis. Please justify this procedure.

What are the units used to quantify FA? ALWAYS give units at all places in the text and in the Figure legends. The numbers in the text on lines 145 to 146 seem completely different than those used in the Figures. In Figures that compare enhancing and decreasing deficiencies it would be much more useful to compare log-fold changes than the raw values. Decreasing fold changes are limited to the 0-1 range, while enhancing have no upper limits.

The data availability statement is misleading. The contents of the Supplementary files available are not made explicit. I do not see any indication that the raw data is available.

Genome-wide screen for deficiencies modifying *Cyclin G*-induced developmental instability in *Drosophila melanogaster*

Valérie Ribeiro¹, Marco Da Costa¹, Delphine Dardalhon-Cuménal¹, Camille A. Dupont¹, Jean-Michel Gibert¹, Emmanuèle Mouchel-Vielh¹, Hélène Thomassin¹, Neel B. Randsholt¹, Vincent Debat^{3*} and Frédérique Peronnet^{1*}

¹ Sorbonne Université, CNRS, Inserm, Développement Adaptation et Vieillessement, Dev2A, F75005 Paris, France

² Sorbonne Université, CNRS, Inserm, Institut de Biologie Paris-Seine, IBPS, F75005 Paris, France

³ Institut de Systématique, Evolution, Biodiversité (ISYEB), Muséum national d'Histoire naturelle, CNRS, Sorbonne Université, EPHE, Université des Antilles CP 50, 57 rue Cuvier, 75005 Paris, France

*** Correspondence:**

Co-last and co-corresponding Author

frederique.peronnet@sorbonne-universite.fr

vincent.debat@mnhn.fr

Running title: Developmental stability

Keywords: Developmental instability, *Drosophila*, Fluctuating asymmetry (FA), Cyclin G, Genetic screen

Reviewer Attachment: August 30, 2025

**ABSTRACT**

Despite long-lasting interest and research efforts, the genetic bases of developmental
stability – the robustness to developmental noise – and its most commonly used estimator,
fluctuating asymmetry (FA), remain poorly understood. The *Drosophila melanogaster*
*Cyclin G* gene (*CycG*) encodes a transcriptional cyclin that regulates growth and the cell
cycle. Over-expression of a potentially more stable isoform of the protein (deleted of a
PEST-rich domain, hereafter called $CycG^{\Delta P}$) induces extreme wing size and shape FA (*i.e.*
high developmental noise), indicating a major disruption of developmental stability. Previous
attempts to identify the genetic bases of FA have been impeded by the constitutively low
level of developmental noise, strongly limiting the power to detect any effect. Here, we
leverage the extreme developmental instability induced by overexpression of $CycG^{\Delta P}$ to
explore the genetic bases of FA: we perform a genome-wide screen for deficiencies that
enhance or reduce $CycG^{\Delta P}$ -induced wing FA. 499 deficiencies uncovering 90% of the
euchromatic genome were combined with a recombinant chromosome expressing $CycG^{\Delta P}$.
We identified 13 and 16 deficiencies that enhance and decrease FA, respectively. Analysis
of mutants for some genes located in these deficiencies shows that Cyclin G ensures
homogeneous growth of organs in synergy with the major morphogens of the wing, Dpp and
Wg, as well as the Hippo and InR/TOR pathways. They also reveal that $CycG^{\Delta P}$ -induced FA
involves Larp, a potential direct interactor of Cyclin G, that regulates translation at the
mitochondrial membrane. This opens up new research perspectives for understanding
developmental stability, suggesting a significant role for mitochondrial activity.

Reviewer Attachment: August 30, 2025

**INTRODUCTION**

Developmental noise results from the inherent variability of biological processes, that
operates independently of genetic or environmental factors (for a review on concepts see
(Debat 2001)). Developmental stability refers to the buffering of developmental noise. In
bilaterians, an estimate of developmental noise is provided by fluctuating asymmetry (FA),
which refers to slight, random deviations from perfect symmetry in paired structures, such
as bilateral organs. The genetic bases of developmental stability remain largely unknown.
Investigation of mandible size and shape FA in a population of mice has identified a few
single-locus Quantitative Loci Traits (QTLs) (Leamy et al. 2015). In *Drosophila*, the wings,
which have a highly stereotyped organisation, are commonly used to measure FA, the
variance in the difference between the left and right wing in a population. Genome-wide
deficiency mapping of regions responsible for developmental noise has identified
deficiencies associated with increased centroid size or shape FA (Takahashi, Okada,
Teramura, et al. 2011; Takahashi, Okada, and Teramura 2011). However, the consistently
low level of developmental noise - which greatly limits the ability to detect any effects - likely
explains why no attempts have been made to identify the genes responsible for these
effects within the QTLs or deficiencies.

In *Drosophila melanogaster*, the *Cyclin G* gene (*CycG*), which encodes a
transcriptional cyclin, has been implicated in developmental noise (Debat et al. 2011; Debat
and Peronnet 2013). Indeed, overexpression of *CycG^{ΔP}*, a modified version of Cyclin G that
lacks the C-terminal PEST domain and is therefore potentially more stable, induces an
unprecedented high level of size and shape FA (40-fold increase compared to control). The
strongly destabilized system induced by *CycG^{ΔP}* overexpression offers a unique opportunity
to investigate the genetic bases of developmental stability. In this study we used *CycG^{ΔP}*
overexpression to perform a genome-wide screen for deficiencies that enhance or reduce
*CycG^{ΔP}*-induced wing size FA. 495 deficiencies uncovering about 90% of the euchromatic
genome were combined with a recombinant chromosome expressing *CycG^{ΔP}*. We thus
identified 13 deficiencies that enhance FA and 16 deficiencies that decrease it. Analysis of
mutants for some genes located in these deficiencies shows that Cyclin G ensures
homogeneous growth of organs in synergy with morphogens and growth control pathways.
They also suggest an impact of mitochondrial activity on developmental stability.

Reviewer Attachment: August 30, 2025

MATERIAL AND METHODS

Genetics

Flies were raised in yeast-cornflour broth (yeast extract 75 g/L, cornflour 90 g/L,
agar-agar 10.7 g/L, methyl 4-hydroxybenzoate 5.5 g/L) at 25°C. Lines carrying deficiencies
(*Df*) were obtained from the Bloomington stock center (Supplementary Table 1). These lines
have been generated in two different *w*¹¹¹⁸ genetic backgrounds (stocks BL-5905 and BL-
6326) that were used as negative controls (further called *w*⁵⁹⁰⁵ and *w*⁶³²⁶, respectively). The
mutant alleles used in this study are listed in Supplementary Table 2. The *da-Gal4,UAS-*
*CycG*^{AP} third chromosome, obtained by genetic recombination (Debat et al. 2011), was
used as a positive control. Due to high lethality and low fertility, this chromosome was
maintained against the *TM6c,Sb* balancer, in males, at 18°C.

In order to address the genetic interactions between *CycG* and each deficiency, 5
replicate crosses containing 6 deficiency or control females and 5 *da-Gal4, UAS-*
*CycG*^{AP}/*TM6c, Sb* or *da-Gal4* males were performed. Series of experiments with up to 25
deficiencies were performed, each series including the appropriate positive and negative
controls. Care was taken to keep the crosses in standardized conditions: tubes used for a
given series were from the same batch, they were maintained at 25°C on the same
incubator floor, parents were transferred into new tubes each 48 h, three times, then
discarded. The offspring was counted and kept in 70 % ethanol. Thirty females of the
desired genotype (*Df/da-Gal4, UAS-CycG*^{AP}, *+/da-Gal4, UAS-CycG*^{AP}, *Df/da-Gal4* or *+/da-*
*Gal4*) were randomly sampled (Supplementary Figure 1).

Measurements and statistical analysis

The wings were mounted and scanned as described (Debat et al. 2011). The
distance between landmarks 3 and 13 was used to estimate wing length (Debat et al. 2011)
(Supplementary Figure 1). Each measurement was taken twice. Genetic interactions
between individual genes and *CycG*^{AP} were tested using the same protocol.

FA of wing length was estimated using the FA10 index as previously described
(Debat et al. 2011). The nortest package was used to test normality of the difference
between left- and right-wing size (Shapiro-Wilk and Lilliefors tests) as well as skewness and
kurtosis. The rare outliers that perturbed normality, skewness (< -0.5 or > +0.5) or kurtosis
(< -3 or > +3) were eliminated from the sample (see number of individuals in Supplementary

Reviewer Attachment: August 30, 2025

Table 3). F-tests were performed to compare FA of the assays to the one of the series
internal controls (*Df/da-Gal4, UAS-CycG^{ΔP}* compared to *+/da-Gal4, UAS-CycG^{ΔP}*, *Df/da-*
*Gal4* compared to *+/da-Gal4*). **Holm-Bonferroni** correction was applied to each series. Inter-
individual variance of wing length represented the variance of the individual mean wing
length in the same population.

RESULTS

Genome-wide screen for deficiencies that modify CycG-induced fluctuating 120 asymmetry

To scan the whole genome for modification of CycG^{ΔP}-induced FA, we used a set of 499
deficiencies uncovering approximately 90% of the euchromatic genome (Supplementary
Table 1). Of these, 4 deficiencies were too weak to be crossed (Supplementary Table 1). A
primary screen was performed by crossing females of the remaining 495 deficiencies with
*da-Gal4, UAS-CycG^{ΔP}/TM6c, Sb* males (Supplementary Figure 1). As the FA values varied
from experiment to experiment, a positive control (*w* females crossed with *da-Gal4, UAS-*
*CycG^{ΔP}/TM6c, Sb* males) and a negative control (*w* females crossed with *da-Gal4/TM6c, Sb*
males) were systematically included in each experimental series to serve as intrinsic
standards. Forty-one deficiencies were lethal or sub-lethal when combined with the *da-*
*Gal4, UAS-CycG^{ΔP}* chromosome, preventing FA analyses (Supplementary Table 1). Two
deficiencies produced damaged wings when combined with *da-Gal4, UAS-CycG^{ΔP}*,
rendered them unmeasurable (Supplementary Table 1). A total of 16,200 fly wings were
mounted and analysed in this primary screen. **Of the 452 deficiencies whose combination**
**with *da-Gal4, UAS-CycG^{ΔP}* produced viable flies with usable wings, most did not modify**
***CycG^{ΔP}*-induced FA (neutral deficiencies).** However, 39 deficiencies and 21 significantly
decreased or increased CycG^{ΔP}-induced FA, respectively (Supplementary Table 1). We
then performed a secondary screen by crossing females of these 60 deficiencies with *da-*
*Gal4, UAS-CycG^{ΔP}/TM6c, Sb* or *da-Gal4* males in order to replicate the primary screen while
measuring the FA of the deficiencies independently of CycG^{ΔP} overexpression. In addition,
8 neutral deficiencies were randomly selected to confirm the absence of an effect. The
results of this secondary screen – a total of 4640 fly wings were mounted and analysed –
are shown in Supplementary Table 3. Of the 68 deficiencies tested, only three,
*Df(2L)BSC17*, *Df(1)BSC582*, and *Df(3L)BSC419*, increased FA when combined with *da-*
*Gal4* in absence of CycG^{ΔP} expression. None decreased it. Nevertheless, the increase was

Reviewer Attachment: August 30, 2025

rather modest compared to that observed in context of *CycG^{ΔP}* expression (3.3, 3.4 and 4.8
 *versus* 28.5, 73.4 and 73.4 for the intrinsic positive controls, respectively). Interestingly,
 *Df(3L)BSC419* partially overlaps *Df(3L)ED4978*, which has already been shown to increase
 wing shape and bristle FA (Takahashi, Okada, Teramura, et al. 2011; Takahashi, Okada,
 and Teramura 2011). The absence of *CycG^{ΔP}*-induced FA modification was verified for the
 8 neutral deficiencies. However, the secondary screen eliminated some deficiencies whose
 effect on *CycG^{ΔP}*-induced FA was no longer significant, probably due to the variability of the
 internal control. Thus, the screen ended with 29 modifier deficiencies: 16 that reproducibly
 decreased *CycG^{ΔP}*-induced FA and 13 that reproducibly enhanced it, further referred to as
 decreasing (D) and enhancing (E) deficiencies, respectively (Table 1 and Figure 1). We did
 not observe any significant correlation between the size of these deficiencies and the
 amplitude of their effect on *CycG^{ΔP}*-induced FA (Figure 2A).

Surprisingly, the 13 E deficiencies clustered on the second chromosome and the left
 arm of the third chromosome whereas the 16 D deficiencies clustered on the X
 chromosome and the right arm of the third chromosome (Table 1). The mean wing size of
 all *Df; da-Gal4, UAS-CycG^{ΔP}* flies was smaller than that of *Df; da-Gal4* flies, indicating that
 *CycG^{ΔP}* expression had a dominant effect on size reduction (Figure 2B and Supplementary
 Figure 2). Lastly, we observed a positive correlation between FA and inter-individual
 variance (Figure 2C). As environmental conditions cannot be fully controlled and individuals
 are not genetically perfectly identical, this inter-individual variance thus reflects minute
 genetic and environmental differences. Whether organisms are simultaneously robust to
 developmental noise, environmental and genetic effects is unclear. This positive correlation
 between FA and inter-individual variance suggests a match between developmental stability
 and environmental/genetic robustness.

Taken together, these results confirmed that *CycG^{ΔP}* expression strongly perturbs
 developmental stability. In addition, we identified 29 deficiencies among the 452 analysed
 that enhanced or decreased *CycG^{ΔP}*-induced FA and could thus contain genes important for
 developmental stability. These genes could therefore be partners of *CycG* in the
 mechanisms that maintain developmental stability.

175 **Identification of genes that modify *CycG^{ΔP}*-induced fluctuating asymmetry**

We next set out to identify partners of *CycG* in developmental stability within these
 deficiencies. Interestingly, *Df(3R)ED6361*, which uncovers *CycG*, reduced the FA induced

Reviewer Attachment: August 30, 2025

by *CycG^{ΔP}* expression, suggesting that expression of *CycG^{ΔP}* behaved as a dominant
negative mutant and that the deviation from symmetry it induced was dose-dependent.
Taken together, the E deficiencies contained 736 genes (576 protein-coding genes and 160
non-protein-coding genes), and the D deficiencies 865 genes (715 protein-coding genes
and 150 non-protein-coding genes), making it impossible to test mutants of these genes
individually. To select which mutants to test, we focused on genes encoding direct protein
partners of Cyclin G or genes involved in the control of growth (Supplementary Table 2).

Although located in D deficiencies (Supplementary Table 3), mutants or RNAi lines
for *Aladin* (Carvalho et al. 2015), *CG9426* (FBgn0032485), *E5* (Dalton et al. 1989), *eEF2*
(Marygold et al. 2017), and *SAK* (Bettencourt-Dias et al. 2004) which encode direct Cyclin
G physical interactors, significantly increased *CycG^{ΔP}*-induced FA (Figure 3). Other genes,
or combinations of genes, in these deficiencies may be responsible for the observed effects.
On the other hand, two RNAi lines for *larp*, which is located in a D deficiency
(*Df(3R)BSC322*) and also encodes a direct Cyclin G physical interactor (11), significantly
reduced *CycG^{ΔP}*-induced FA (Figure 3). *Larp* and Cyclin G may thus be direct partners in a
process important for developmental stability.

The Hippo signalling pathway plays a key role in controlling organ size (10). *Merlin*
(*Mer*), which encodes a positive regulator of this pathway that interacts directly with Cyclin
G (11), is located in the D deficiency *Df(1)BSC871*. Additionally, *hippo* (*hpo*), which
encodes the kinase of the pathway, is located in *Df(2R)ED3728*, which increased *CycG^{ΔP}*-
induced FA. This prompted us to investigate the impact of other members of the Hippo
pathway on *CycG^{ΔP}*-induced FA. Mutants for *Mer*, *hpo*, *ft*, *exp*, *wts* and *ban*, significantly
increased it (Figure 4A). Furthermore, *Mer* inactivation also increased FA independently of
*CycG* deregulation (Figure 4A). These results suggest that the Hippo pathway plays a role
in developmental stability.

The *mTor* gene, which encodes the central member of another key regulator of
growth (Grewal 2009), the InsulinR/TOR signalling pathway (InR/TOR) belongs to a
deficiency that reduced *CycG^{ΔP}*-induced FA [*Df(2L)BSC277*]. Furthermore, Cyclin G
interacts directly with three actors of this pathway : Akt1, PP2A and Wdb (Fischer et al.
2016; Oughtred et al. 2021). We thus tested the effect of *mTor* and other members of the
pathway for their effect on *CycG^{ΔP}*-induced FA. Most of these, particularly mutants for *Akt1*,
*S6K*, *foxo*, *PTEN*, *mTor* and *wdb*, increased *CycG^{ΔP}*-induced FA. This suggests that the
interaction between Cyclin G and the InR/TOR pathway is crucial for developmental stability

Reviewer Attachment: August 30, 2025

(Figure 4B).

Finally, two major morphogens of the wing, *dpp* and *wg* [for a review see (Tabata and
Takei 2004)], belonged to deficiencies that enhanced *CycG^{ΔP}*-induced FA [*Df(2L)Exel7011*
and *Df(2L)BSC291*, respectively]. Mutants for both genes also increased *CycG^{ΔP}*-induced
FA, indicating that we had identified at least one gene responsible for this effect within each
deficiency (Figure 4C).

DISCUSSION

In our efforts to identify deficiencies that modify *CycG^{ΔP}*-induced FA, we found that
13 deficiencies out of 452 enhanced it (E deficiencies) while 16 deficiencies decreased it (D
deficiencies). Notably, these deficiencies were not evenly distributed across all
chromosomes. The D deficiencies were enriched on chromosomes X and 3R, whereas the
E deficiencies were concentrated on chromosomes 2 and 3L. This pattern suggests that
chromosomes 2 and 3L may have a general positive effect on developmental stability. In
addition, we observed that the vast majority of modifier deficiencies (65 out of the 68 tested)
does not affect FA on their own. Finally, the 13 deficiencies that enhance *CycG^{ΔP}*-induced
FA represent only 4.1 % of the *Drosophila melanogaster* genome. Collectively, these results
indicate that under normal rearing conditions the control of growth is highly robust.

Overexpression of *CycG^{ΔP}* using the Gal4/UAS system provides an entry point to
discover genes and gene networks that are essential for growth homeostasis. Not
surprisingly, we found that the major morphogens (**Dpp** and *Wg*) and growth pathways
(InR/Tor and Hippo pathways) are essential for developmental stability. In line with our
findings, alternative splicing of *yki*, which encodes the transcription factor Yorkie, a target of
the Hippo pathway, has previously been shown to contribute to developmental stability
(Srivastava et al. 2020 Dec 15). Furthermore, Merlin that promotes assembly of a functional
Hippo signaling complex at the apical cell cortex has been shown to directly interact with
Cyclin G (Oughtred et al. 2021). Beyond its specific role, Cyclin G may be unique in that it
sits at the crossroads of many signalling pathways. Our previous systems biology approach
has indeed shown that *CycG* is linked to many genes in the wing imaginal disc (Dardalhon-
Cuménalet al. 2018). These new findings confirm that *CycG* may be a hub within a
complex genetic network important for the robustness of organ growth. We have previously

Reviewer Attachment: August 30, 2025

shown that expression of *CycG^{ΔP}* abolishes the strong negative correlation between cell
size and cell number observed in wild-type fly wings (Debat et al. 2011). Cyclin G may
orchestrate growth homeostasis by ensuring the proper coordination of growth and the cell
cycle, which would be critical for developmental stability. This central role may explain the
spectacular effects of its deregulation on fluctuating asymmetry.

Some potential direct Cyclin G partners are included in the deficiencies that modify
*CycG^{ΔP}*-induced FA. Strikingly, these partners belong to different gene ontology categories,
ranging from the aforementioned Merlin to transcription factors (e.g. E5), translation
regulators (e.g. *Larp*) and proteins involved in protein degradation (e.g. *Cul2*), reinforcing
the idea that Cyclin G may be involved in diverse cellular processes. Among the genes
encoding these potential direct partners, *lar*p is the only one whose inactivation strongly
reduces *CycG^{ΔP}*-induced FA, suggesting that the effect of Cyclin G on FA requires this
direct interaction. *Larp* encodes an evolutionarily conserved RNA-binding protein that forms
a complex with the Poly(A) binding protein (PABP), a translation regulator (Blagden et al.
2009; Deragon and Bousquet- Antonelli 2015). In addition, *Larp* recognises mRNAs with a
5' Terminal OligoPyrimidine (5' TOP) motif, i.e. mRNAs encoding proteins essential for
protein synthesis such as ribosomal proteins or translation factors (Cockman et al. 2020).
More recently, *Larp* has been shown to play a dual role as a translational repressor and a
stabiliser of 5' TOP mRNAs (20). In particular, when phosphorylated by Pink1, *Larp* inhibits
translation at the mitochondrial outer membrane suggesting that it may have an effect on
the production of new mitochondria (Zhang et al. 2019). Interestingly, in wing imaginal discs
expressing *CycG^{ΔP}*, genes involved in translation are up-regulated and those involved in
metabolism and mitochondrial activity are down-regulated (Dardalhon-Cuménal et al. 2018).
Taken together, these data support the idea that there may be a link between
developmental stability and the regulation of mitochondrial activity.

FIGURES

**Figure 1: Secondary screen – Analysis of the 60 candidate deficiencies isolated from**
**the primary screen.**

**a** – FA of 39 deficiencies that decreased *CycG^{ΔP}*-induced FA in the primary screen

Reviewer Attachment: August 30, 2025

normalised to the FA of *da-Gal4, UAS-CycG^{ΔP}* (red line). 16 of them reproduced the results
of the primary screen (orange).

**b** – FA of 21 deficiencies that enhanced *CycG^{ΔP}*-induced FA in the primary screen
normalised to the FA of *da-Gal4, UAS-CycG^{ΔP}* (red line). 13 of them reproduced the results
of the primary screen (green).

*p<0.05, **p<0.01, ***p<0.001

**Figure 2: Secondary screen - Relationship between fluctuating asymmetry and**
**deficiency length, mean wing size and inter-individual variance.**

**a** – FA (fold change) of the 16 decreasing (D) deficiencies (orange), the 13 enhancing (E)
deficiencies (green) and 8 neutral deficiencies (grey) versus deficiency length. No
significant correlation was observed between the two parameters (Pearson coefficient, R = -
0.15).

**b** - FA of the 16 decreasing (D) deficiencies (orange) and the 13 enhancing (E) deficiencies
(green) combined with *da-Gal4, UAS-CycG^{ΔP}* (filled circles) or not (empty circles) versus
mean wing size. Filled grey square: *+/da-Gal4, UAS-CycG^{ΔP}*; empty grey square: *+/da-*
*Gal4*. A strong negative correlation was observed between these parameters (R = -0.65).

**c** – FA (log10) of the 16 decreasing (D) deficiencies (orange) and the 13 enhancing (E)
deficiencies (green) combined with *da-Gal4, UAS-CycG^{ΔP}* (filled circles) or not (empty
circles) versus inter-individual variance (log10). Filled grey square: *+/da-Gal4, UAS-CycG^{ΔP}*;
empty grey square: *+/da-Gal4*. A strong positive correlation was observed between the two
parameters (R = +0.67).

**Figure 3: Candidate gene analyses – Genes encoding direct protein partners of**
**Cyclin G located in deficiencies that modify *CycG^{ΔP}*-induced FA.**

In green, alleles that significantly increased *CycG^{ΔP}*-induced FA. In orange, alleles that
significantly decreased *CycG^{ΔP}*-induced FA. In light grey, positive control (*+/da-Gal4, UAS-*
*CycG^{ΔP}*). In dark grey, alleles that did not modify *CycG^{ΔP}*-induced FA. Several alleles were
lethal (†) when combined to *+/da-Gal4* or *+/da-Gal4, UAS-CycG^{ΔP}*.

*p<0.05, **p<0.01, ***p<0.001

**Figure 4: Candidate gene analyses – Genes encoding members of the Hippo pathway**

Reviewer Attachment: August 30, 2025

**(a), the InR/TOR pathway (b) and the Dpp and Wg morphogens (c).**

In green, alleles that significantly increased *CycG*^{ΔP}-induced FA. In grey, alleles that did not
 modify *CycG*^{ΔP}-induced FA.

*p<0.05, **p<0.01, ***p<0.001

**Supplementary Figure 1:**

**a** - Position of the 15 landmarks on the wing. Landmarks 3 and 13 (red) were used to
 measure wing length.

**b** – Diagrams of primary and secondary screens.

*Df*: deficiency; *w*ⁿ: *w*⁵⁹⁰⁵ or *w*⁶³²⁶; *Bal*: balancer chromosome; N: neutral deficiencies; D:
 deficiency that decreased *CycG*^{ΔP}-induced fluctuating asymmetry; E: deficiency that
 increased *CycG*^{ΔP}-induced fluctuating asymmetry.

**Supplementary Figure 2: Secondary screen – Mean wing size of flies heterozygous**
 **for the deficiencies associated with *da-Gal4* (top) or *da-Gal4, UAS-CycG*^{ΔP} (down).**

Box-plots showing mean wing size. In grey, mean wing size of *+/da-Gal4* (**a**) or *+/da-*
 *Gal4, UAS-CycG*^{ΔP} flies. In orange, mean wing size of deficiencies that decreased *CycG*^{ΔP}-
 induced FA combined with *da-Gal4* (top) or *da-Gal4, UAS-CycG*^{ΔP} (down). In green, mean
 wing size of deficiencies that increased *CycG*^{ΔP}-induced FA combined with *da-Gal4* (top) or
 *da-Gal4, UAS-CycG*^{ΔP} (**b**).

The two red lines correspond to the mean wing size of *+/da-Gal4* (1161) and *+/da-Gal4,*
 *UAS-CycG*^{ΔP} (1014) flies.

*p<0.05, **p<0.01, ***p<0.001

Reviewer Attachment: August 30, 2025

**Table 1: Deficiencies modifying CycG^{ΔP}-induced FA.**

The fluctuating asymmetry fold-change corresponds to the ratio between fluctuating
 asymmetry of the deficiency combined to *da-Gal4, UAS-CycG^{ΔP}* and fluctuating asymmetry
 of *da-Gal4, UAS-CycG^{ΔP}*.

Chromosome arm	Deficiency name	Cytological location	Fluctuating asymmetry fold-change	Adjusted p-value
X	Df(1)BSC530	1A5;1B12	0.37329	4.68E-02
	Df(1)M38-C5	8B;8E	0.24024	2.86E-03
	Df(1)BSC722	10B3;10E1	0.25570	6.72E-03
	Df(1)FDD-0024486	14C4;14D1	0.12722	2.58E-06
	Df(1)BSC643	15F9;16F1	0.28609	7.33E-03
	Df(1)BSC405	16D5;16F6	0.37682	4.68E-02
	Df(1)BSC871	18D7;18F2	0.32513	2.55E-02
2L	Df(2L)Exel7011	22E1;22F3	3.67005	4.34E-03
	Df(2L)BSC354	26D7;26E3	4.32949	1.32E-03
	Df(2L)BSC291	27D6;27F2	2.69177	3.96E-02
	Df(2L)Exel7034	28E1;28F1	3.92670	2.69E-03
	Df(2L)ED629	29B4;29E4	6.02764	5.81E-05
	Df(2L)BSC277	34A1;34B2	0.39515	2.82E-02
	Df(2L)ED1473	39B4;40A5	3.37401	7.77E-03
	Df(2L)lt109	h35;h35	2.87894	2.34E-02
2R	Df(2R)BSC880	49A9;49E1	3.36810	9.85E-04
	Df(2R)ED3728	56D10;56E2	2.79286	2.50E-02
3L	Df(3L)ED201	61B1;61C1	9.14411	6.11E-07
	Df(3L)ED4457	67E2;68A7	3.77715	3.03E-03
	Df(3L)BSC12	69F6;70A2	7.46921	5.43E-06
	Df(3L)ED4710	74D1;75B11	5.14760	1.95E-04
	Df(3L)BSC419	78C2;78D8	0.20077	5.30E-04
3R	Df(3R)BSC549	83A6;83B6	0.28288	9.98E-03
	Df(3R)ED5623	87E3;88A4	0.32086	1.65E-02
	Df(3R)ED5705	88E12;89A5	0.33282	1.69E-02
	Df(3R)ED6255	97D2;97F1	0.27386	5.53E-03
	Df(3R)BSC322	98C3;98D7	0.33719	3.21E-02
	Df(3R)BSC749	100B1;100C1	0.39540	9.04E-03
	Df(3R)ED6361	100C7;100E3	0.40559	1.17E-02

Reviewer Attachment: August 30, 2025

**Supplementary Table 1: Deficiencies used in this study.**

**Supplementary Table 2: Alleles used in this study to identify genes that modify**

***CycG*^{ΔP}-induced fluctuating asymmetry.**

**Supplementary Table 3: Secondary screen for deficiencies which modify *CycG*^{ΔP}-**

**induced FA.**

Reviewer Attachment: August 30, 2025

**Data availability statement**

The authors confirm that all the data necessary to verify the conclusions of the article are
included in the text, figures and tables. Strains are available upon request.

**Acknowledgments**

The authors would like to thank all the members of the Heterochromatin, Cell Fate and
Exposome team for their valuable advice and productive discussions. They would also like
to thank Dr. Y. Bellaïche for providing the *Akt1[1]* mutant, and the Bloomington Stock
Center for mutant and deficiency strains.

**Study funding**

This study was funded by ongoing support from the Centre National de la Recherche
Scientifique (CNRS) and Sorbonne Université (SU).

Reviewer Attachment: August 30, 2025

**REFERENCES**

- Debat V. 2001. Mapping phenotypes: canalization, plasticity and developmental stability.
Trends in Ecology & Evolution. 2001;16:555–561.
- Leamy L et al. 2015. The Genetic Architecture of Fluctuating Asymmetry of Mandible Size
and Shape in a Population of Mice: Another Look. Symmetry. 2015;7(1):146–163.
<https://doi.org/10.3390/sym7010146>
- Takahashi KH, Okada Y, Teramura K, Tsujino M. 2011. Deficiency mapping of the genomic
regions associated with effects on developmental stability in *Drosophila melanogaster*.
Evolution. 2011;65:3565–3577.
- Takahashi KH, Okada Y, Teramura K. 2011. Genome-Wide Deficiency Mapping of the
Regions Responsible for Temporal Canalization of the Developmental Processes of
*Drosophila melanogaster*. J Hered. 2011;102(4):448–57.
<https://doi.org/10.1093/jhered/esr026>
- Debat V et al. 2011. Developmental Stability: A Major Role for Cyclin G in *Drosophila*
*melanogaster*. PLoS Genet. 2011;7(10):e1002314.
<https://doi.org/10.1371/journal.pgen.1002314>
- Debat V, Peronnet F. 2013. Asymmetric flies: The control of developmental noise in
*Drosophila*. Fly (Austin). 2013;7(2):1–8. <https://doi.org/10.4161/fly.23558>
- Carvalhal S et al. 2015. The nucleoporin ALADIN regulates Aurora A localization to ensure
robust mitotic spindle formation Zheng Y, editor. Molecular Biology of the Cell. 2015
[accessed 2025 Jun 5];26(19):3424–3438.
<https://www.molbiolcell.org/doi/10.1091/mbc.E15-02-0113>. <https://doi.org/10.1091/mbc.E15-02-0113>
- Dalton D, Chadwick R, McGinnis W. 1989. Expression and embryonic function of empty
spiracles: a *Drosophila* homeo box gene with two patterning functions on the anterior-
posterior axis of the embryo. Genes & Development. 1989 [accessed 2024 Nov
11];3(12a):1940–1956. <http://www.genesdev.org/cgi/doi/10.1101/gad.3.12a.1940>.
<https://doi.org/10.1101/gad.3.12a.1940>
- Marygold SJ, Attrill H, Lasko P. 2017. The translation factors of *Drosophila melanogaster*.
Fly (Austin). 2017;11(1):65–74. <https://doi.org/10.1080/19336934.2016.1220464>
- Bettencourt-Dias M et al. 2004. Genome-wide survey of protein kinases required for cell
cycle progression. Nature. 2004;432(7020):980–7. <https://doi.org/10.1038/nature03160>
- Grewal SS. 2009. Insulin/TOR signaling in growth and homeostasis: a view from the fly
world. Int J Biochem Cell Biol. 2009;41(5):1006–10.
<https://doi.org/10.1016/j.biocel.2008.10.010>
- Fischer P, Preiss A, Nagel AC. 2016. A triangular connection between Cyclin G, PP2A and
Akt1 in the regulation of growth and metabolism in *Drosophila*. Fly. 2016:1–8.
<https://doi.org/10.1080/19336934.2016.1162362>

Reviewer Attachment: August 30, 2025

- Oughtred R et al. 2021. The BioGRID database: A comprehensive biomedical resource of
curated protein, genetic, and chemical interactions. *Protein Science*. 2021 [accessed 2024
Nov 2];30(1):187–200. <https://onlinelibrary.wiley.com/doi/10.1002/pro.3978>.
<https://doi.org/10.1002/pro.3978>
- Tabata T, Takei Y. 2004. Morphogens, their identification and regulation. *Development*.
2004 [accessed 2024 Nov 11];131(4):703–712.
[https://journals.biologists.com/dev/article/131/4/703/42586/Morphogens-their-identification-](https://journals.biologists.com/dev/article/131/4/703/42586/Morphogens-their-identification-and-regulation)
[and-regulation. https://doi.org/10.1242/dev.01043](https://doi.org/10.1242/dev.01043)
- Srivastava D et al. 2020. Modulation of Yorkie activity by alternative splicing is required for
developmental stability. *The EMBO Journal*. 2020 Dec 15 [accessed 2020 Dec 28].
<https://onlinelibrary.wiley.com/doi/10.15252/embj.2020104895>.
<https://doi.org/10.15252/embj.2020104895>
- Dardalhon-Cuménal D et al. 2018. Cyclin G and the Polycomb Repressive complexes
PRC1 and PR-DUB cooperate for developmental stability. *PLoS Genet*.
2018;14(7):e1007498. <https://doi.org/10.1371/journal.pgen.1007498>
- Blagden SP et al. 2009. Drosophila Larp associates with poly(A)-binding protein and is
required for male fertility and syncytial embryo development. *Developmental Biology*. 2009
[accessed 2025 Feb 9];334(1):186–197.
<https://linkinghub.elsevier.com/retrieve/pii/S0012160609010665>.
<https://doi.org/10.1016/j.ydbio.2009.07.016>
- Deragon J, Bousquet-Antonelli C. 2015. The role of LARP1 in translation and beyond.
*WIREs RNA*. 2015 [accessed 2024 Nov 11];6(4):399–417.
<https://wires.onlinelibrary.wiley.com/doi/10.1002/wrna.1282>.
<https://doi.org/10.1002/wrna.1282>
- Cockman E, Anderson P, Ivanov P. 2020. TOP mRNPs: Molecular Mechanisms and
Principles of Regulation. *Biomolecules*. 2020 [accessed 2025 Feb 9];10(7):969.
<https://www.mdpi.com/2218-273X/10/7/969>. <https://doi.org/10.3390/biom10070969>
- Berman AJ et al. 2021. Controversies around the function of LARP1. *RNA Biology*. 2021
[accessed 2024 Nov 11];18(2):207–217.
<https://www.tandfonline.com/doi/full/10.1080/15476286.2020.1733787>.
<https://doi.org/10.1080/15476286.2020.1733787>
- Zhang Y et al. 2019. PINK1 Inhibits Local Protein Synthesis to Limit Transmission of
Deleterious Mitochondrial DNA Mutations. *Molecular Cell*. 2019 [accessed 2022 Jun
6];73(6):1127–1137.e5. <https://linkinghub.elsevier.com/retrieve/pii/S1097276519300139>.
<https://doi.org/10.1016/j.molcel.2019.01.013>

Figure 1

Figure 2

Figure 3

Figure 4

Rebuttal Letter

Reviewer #1

Please note I have also provided detailed comments on the PDF (notes associated with each highlighted segment of text).

This study was clearly a substantial body of work. Using FA to study mechanisms modulating developmental stability is really hard, requiring very careful experiments, large sample sizes, analyses and repeated measurements of samples. Despite a few issues I will discuss in detail below, the authors of this study managed to do a good job on most of these counts, so kudos! I think in principle, that once some important revisions are made, and a few additional analyses are performed to check for some potential confounding effects (in particular for cases where the modifiers of the CycG perturbation themselves impact wing size, and how this can influence the FA of the wing), this will be a valuable contribution to the genetic analysis of developmental stability in Drosophila.

Having said all of this I have a few major concerns that I am highlighting for the editor consideration in evaluating this manuscript for suitability for publication in Genetics.

We would like to thank the reviewer for his careful proofreading of our manuscript and for his constructive feedback. We believe that this new version, which takes his remarks into account, is significantly improved.

*One major issue is both context and scholarship. As I note in my direct comments on the PDF, this is not the first study using deletion mapping to identify regions that influence the magnitude of FA (the authors briefly refer to some of those studies in the results in discussion of one of the deficiencies which influences FA in the absence of the CycG perturbation). There are also a number of other studies examining specific genes and underlying biological processes and how they influence FA (and presumably developmental stability). Not just the genes that the co-senior author Dr. Debat (which I am a co-author on with Dr. Debat, so feel free to not add that as a citation) has examined (Debat et al. 2009). But also the influence of perturbation in genes like *dllp8*, *lgr3*, *Tps1*, *Hsp67*, *Yorkie* and I am sure other ones in the literature I am not aware of. So what is fundamentally new in this study is using the CycG perturbation (the over-expression of the stable version of CycG) as the basis to perform a modifier screen with deficiency mapping. So I really think the work in this study needs to be put in the larger biological context of what is known for this (in the introduction and in discussing the results of the current study). Indeed, especially because the very odd nature of the CycG perturbation (i.e. over-expression of a very stable form CycG) it is worth putting how this perturbation increases developmental instability as compared to some of the other mechanisms (in particular the global aspects for both the *dllp8/lgr3* and the *trehalose/Tps1* side of things).*

We added a paragraph to the introduction to present the other genes involved in developmental stability. In the discussion section, we further explore the links between Cyclin G and the other mechanisms that contribute to developmental stability (i.e. the BMP pathway, the *Dilp8/Lgr3* axis, and *Tsp1*).

See lines 64-81 in the introduction, lines 286-311 in the discussion.

My second major issue is the incredible brevity of the material and methods. I had to back track to several other studies to figure out what was going on, and also do some sleuthing in the supplemental tables of the current manuscript. All of the specific issues are documented on the PDF. Please include a reasonable amount of detail so that readers can focus on reading study to understand what you did. As it turns out, what you did (after I did my sleuthing) in terms of experimental designs, controls (mostly), sampling, repeated measuring and choice of measure for FA are good, so make that all clear to the readers so they can see that.

We sincerely apologise for the lack of detail in the Materials and Methods section, which may have made it difficult for the reviewer to evaluate our work. We have now added the necessary information to this section.

See lines 130-155.

My only major concern methodologically has to do with some of the deletions including in this modifier screen. Most are from either the DrosDel or Exelixis deletion collections, and it seems like the authors used the corresponding co-isogenic control lines for each deletion collection as appropriate. However, some of the deletions are deficiencies that do not belong to either collection (but are available at the stock centre nonetheless). So in these cases, it is not clear what the control strains would be. So I think these strains need to be identified and highlighted as having potential confounds of both the hemizygous allelic effects uncovered by the deletion, but also a distinct genetic background for which there is no corresponding co-isogenic control strain. It may be that these still uncover really interesting modifiers, but it also may be that these effects are epistatic with the larger genetic background effects of these orphan deletion strains. I think it is a relatively modest (10%?) of the deletion strains you used, but clarifying all of this, and providing the appropriate caveats will be very important.

This point has been clarified. Deletions with an unknown background are now indicated in Supplementary Figure 1. These represent 59 deficiencies out of the 499 analysed deficiencies (11.8%). This information has been added to the Materials and Methods section, and a corresponding comment has been included in the Results.

See lines 104-113 in the Materials and Methods section and lines 168-172 in the Results section.

Did you use FA10a or FA10b (I didn't name these things). Also add a citation to the 2003 Palmer and Strobeck book chapter. P&S 86 is a classic, but the book chapter is much clearer for people coming into the field.

We used untransformed values, so FA10a. We have expanded the Material and Methods section so at to make it more self-explicit. We apologise for this lack of details in the previous version.

See lines 130-155.

I highly suggest a density plot or histogram of the distributions (for the various controls, including the CycG perturbations without the deletions) of FAs to give us a better sense of what it looked like for the primary and secondary screen. How extreme are the ones you focused on? These plots will help the readers a great deal.

We have added histograms showing the FA distributions for both the primary and secondary screens.

See Supplementary Figure 2.

Given that Kazuo Takahashi did the deletion screens previously, why not do a more formal comparison of what modifiers you identified and what regions they uncovered in there numerous studies on this? Could be really interesting.

Takahashi et al. (2011) used deficiencies covering around 65% of the genome. Several of these overlapped or were shared with the 29 deficiencies that modify CycG^{ΔP}-induced FA. However, none of these deficiencies show a significant increase in wing centroid size FA in females. This highlights the power of CycG^{ΔP} to reveal genes or genomic regions involved in developmental stability. This information has been added to the text.

See lines 203-207 and Supplementary Table 4.

Depending on which version of FA10 you used (see above), you may still have to contend with the dreaded effects of trait size vs trait asymmetry magnitudes issues. Examining the relationships between your FA10 measure and how much smaller or larger the mean trait size gets for those modifiers (relative to the reduction that the CycG perturbation causes on its own). Hopefully little association, but this needs to be examined empirically. Obviously logL - logR really helps with this issue, but does not make it all go away.

1st Revision - Authors' Response to Reviewers: November 11, 2025

The mean wing size of all *Df; da-Gal4, UAS-CycG^{ΔP}* flies was smaller than that of *Df; da-Gal4* flies. This indicates that *CycG^{ΔP}* expression had a dominant effect on size reduction. However, these differences in asymmetry were not due to a size-dependent effect, as the most asymmetric genotypes had smaller wings (and not larger) and thus reflected genuine differences in developmental stability.

See lines 208 to 213.

Additional comments in the pdf:

However, the consistently low level of developmental noise - which greatly limits the ability to detect any effects - likely explains why no attempts have been made to identify the genes responsible for these effects within the QTLs or deficiencies.

Comment: This is a guess. I am not sure this is actually the case. After all, one of the senior authors showed clearly that heterozygote mutants varied for degree of FA. It just requires high precision measurement and good sample sizes.

We agree with the reviewer and have therefore removed this sentence.

Whether organisms are simultaneously robust to developmental noise, environmental and genetic effects is unclear.

Comment: But this has been examined extensively for this exact trait, so I think you can speak to this in more depth.

We are now discussing the relationship between inter-individual variance and fluctuating asymmetry. See lines 93-98 and lines 214-224.

Comments: I think this is formally either called the Holm correction or the sequential Bonferroni.

We have replaced Holm-Bonferroni by Holm correction.

See line 153.

Individuals are not genetically perfectly identical.

Comments: Do you mean due to new mutations or segregating variation in the parental lines (or both)?

This point was clarified.

See line 219.

We have replaced the D and E deficiencies by decreasing and enhancing deficiency, respectively, throughout the text.

Reviewer #2

This manuscript reports a useful screen for deficiencies and genes that interact with a fluctuating asymmetry inducing mutation, CycG^{DP} to change levels of FA. The results show that quite a few regions affect FA, and that major morphogen loci within those deficiencies potentially contribute to this. In addition, mutations or knockdowns of genes in the major developmental signaling pathways outside of the deficiencies interacting with CycG^{ΔP} also interacted with CycG^{ΔP} to affect FA. Altogether the results are an interesting compilation of candidates for genes potentially responsible for variation in FA.

We would like to thank the reviewer for his positive and constructive comments on our manuscript. Our detailed, point-by-point responses are presented below. We have revised the text accordingly and are confident that it has been significantly improved.

There are two issues raised by the results that the manuscript does not mention that I think would enhance the manuscript. The first is that the results suggest that variation in FA can arise through any of the genetic processes that affect growth - at least there is no pathway the included in the experiments that did not affect FA in this background with heightened sensitivity. This supports the idea that FA is likely to be highly polygenic. This has implications for the mechanisms underlying FA, which seem unlikely to be the result of some master regulatory process.

This point is now emphasised in the introduction (see lines 78-82).

The second is that it remains unclear whether CycG is the only way to globally destabilize the phenotype. This is not a criticism of the experiments - they were not designed to address this question. However, it is of interest to realize that this is not necessarily unique in this way. We just do not know.

CycG^{ΔP} is clearly not the only way to destabilise the phenotype. We have added a paragraph to the introduction to present the other genes involved in developmental stability (see lines 70-78). Our previous results indicate that CycG may be a hub within a complex genetic network that is crucial for the robustness of organ growth (see lines 315-316).

The statistical methods are not really explained. Please explain the FA10 index, and why it was chosen over alternatives. The wing lengths were each measured twice, but how were these repeated measures incorporated into the analysis? How repeatable were the lengths? If repeatability is high, it might be acceptable to simply average measurements, but not if it is low. What is a 'series', in other words what group of results was the Hom-Bonferroni corrections applied to? F ratios of what exactly were used to test for differences in FA? The outlier removal procedure is ad hoc, depending on effects on summary statistics about the distribution assuming that normality is an appropriate null hypothesis. Please justify this procedure.

We sincerely apologise for the lack of detail in the Materials and Methods section, which may have made it difficult for the reviewer to evaluate our work. We have now added the necessary information to this section.

See lines 130-155.

What are the units used to quantify FA? ALWAYS give units at all places in the text and in the Figure legends.

As FA10 is derived from variances (as now explained in the main text line 140), it is thus expressed in squared units of measurement, here squared mm. Units are usually not provided in studies of FA, as they are not easy to interpret for most indices (see Palmer 1994, Fluctuating asymmetry analysis: a primer. In: Developmental Instability: Its Origins and Evolutionary Implications. T. A. Markow. Netherlands, Kluwer Academic Publishers; p 335–364).

1st Revision - Authors' Response to Reviewers: November 11, 2025

The numbers in the text on lines 145 to 146 seem completely different than those used in the Figures. In Figures that compare enhancing and decreasing deficiencies it would be much more useful to compare log-fold changes than the raw values. Decreasing fold changes are limited to the 0-1 range, while enhancing have no upper limits.

Figure 1 now indicates in the axis titles that we are comparing the fold-changes of FA [*i.e.* the ratio between the FA of each deficiency combined with *da-Gal4*, *UAS-CycG^{ΔP}* to the FA of the positive control (*da-Gal4*, *UAS-CycG^{ΔP}*) in each series]. However, we have kept our initial choice to use raw data rather than log-fold changes, as we believe that untransformed data provides a clearer understanding of the results.

In the text, we have compared the ratio of the FA of each deficiency (*Df*; *+/da-Gal4*) to the FA of the negative control of the same series (*+/da-Gal4*), and the ratio of the positive control of the same series (*+/da-Gal4*, *UAS-CycG^{ΔP}*) to the negative control of the same series (*+/da-Gal4*). This shows that the amplitude generated by the deficiencies alone is much lower than that generated by *CycG^{ΔP}*. This point has been made clearer in the text.

See lines 188-191.

The data availability statement is misleading. The contents of the Supplementary files available are not made explicit. I do not see any indication that the raw data is available.

The raw data have been deposited on the Zenodo platform.

See Data availability statement lines 437-440.

December 10, 2025

RE: GENETICS-2025-308768

Dear Dr. Peronnet:

I am pleased to accept your manuscript titled "Genome-wide screen for deficiencies modifying Cyclin G-induced developmental instability in *Drosophila melanogaster*." for publication in GENETICS, pending minor revision.

Please submit your revision along with a brief description of how you modified the manuscript in response to the reviewers' concerns and suggestions (which can be viewed at the bottom of this email). In addition, I read the manuscript carefully and provide a few minor edits on the attached PDF. I expect you should be able to submit a revised manuscript within 30 days. A suitably revised manuscript will be acceptable for publication; I don't expect to send it out for review.

When revising the ms., please make an effort to shorten it, because that almost always improves a manuscript. We urge authors to heed the advice of Strunk and White: "omit needless words"¹. Follow this link to submit the revised manuscript: Link Not Available

Thank you for submitting this story to Genetics.

Sincerely,

Catherine Peichel
Associate Editor
GENETICS

Approved by:
David Greenstein
Senior Editor
GENETICS

Reviewer comments:

Reviewer #1 :

Thank you for making all of the revision. I think the revised version is in great shape and will be of interest to readers of Genetics (like myself). I have a very small number of points that should be trivial to fix.

For the data availability statement, the data is on zenodo (like the authors mention), but the link was not working for me (all I had to do was go to zenodo and search the title of the paper). The authors do include the doi, but I suggest using the full zenodo url to make it easier to find. (<https://doi.org/10.5281/zenodo.17416430>). Also if it is not too much trouble, add the R analysis scripts along with the text files containing all of the size information. I think this will be a useful data set that people will be interested in exploring.

In terms of FA10a and the size effects, I mostly agree with the authors on this. Nonetheless, I think the statement on lines 209-214 of the revised manuscript is a bit strong, and maybe confusing in light of figure 2B. Maybe revise this (the lines on 209-214) to point out that while scale can be an issue, usually this results in increasing sd/variances (and thus potentially FA the way it is measured here) with increasing trait size. In this work you see the opposite (the smallest wings, i.e. those most perturbed by the combination of the *cycG*[deltaP] and specific deletions) most often show the greatest increase in FA and inter-individual variance. Thus the results are not consistent with the expected pattern of scaling effects. Most likely reflecting the impact of increasing perturbations on the developmental system.

If it is easy to compute (and it is possible no one has bothered to figure out the standard error), confidence intervals on the fold changes in FA would be great at least in the table (in addition to the estimate and the Holm adjusted p-value). If no estimator for SE of FA10 is available for this to aid in this calculation, (and I understand not wanting to go back to do the non-parametric bootstrap for all of this), at least mention why it was not included.

The first time you use the terms "decreasing deficiencies" or "increasing deficiencies", just clarify you mean these in terms of

"deletions that decrease FA". I know it seems obvious, but when I was re-reading it, it did take me a bit as it being used as a proper noun, but it does not read that way.

Associate Editor comments:
Please see the attached PDF.

Answers to the reviewer

We thank the reviewer for his comments and careful reading of the manuscript.

Our answers follow.

For the data availability statement, the data is on zenodo (like the authors mention), but the link was not working for me (all I had to do was go to zenodo and search the title of the paper). The authors do include the doi, but I suggest using the full zenodo url to make it easier to find. (<https://doi.org/10.5281/zenodo.17416430>). Also if it is not too much trouble, add the R analysis scripts along with the text files containing all of the size information. I think this will be a useful data set that people will be interested in exploring.

Done. See line 441 of the Word file

*In terms of FA10a and the size effects, I mostly agree with the authors on this. Nonetheless, I think the statement on lines 209-214 of the revised manuscript is a bit strong, and maybe confusing in light of figure 2B. Maybe revise this (the lines on 209-214) to point out that while scale can be an issue, usually this results in increasing sd/variances (and thus potentially FA the way it is measured here) with increasing trait size. In this work you see the opposite (the smallest wings, i.e. those most perturbed by the combination of the *cycG*[ΔP] and specific deletions) most often show the greatest increase in FA and inter-individual variance. Thus the results are not consistent with the expected pattern of scaling effects. Most likely reflecting the impact of increasing perturbations on the developmental system.*

We thank the reviewer for this comment, and have removed the sentence “indicating that *CycG* ^{ΔP} expression had a dominant effect on size reduction”. However, we did not add the statement that “size reduction reflects the impact of increasing perturbations on the developmental system”, as this is already implied by the phrase “thus reflected genuine differences in developmental stability (Palmer, A. R. and Strobeck, C. 2003).”

If it is easy to compute (and it is possible no one has bothered to figure out the standard error), confidence intervals on the fold changes in FA would be great at least in the table (in addition to the estimate and the Holm adjusted p-value). If no estimator for SE of FA10 is available for this to aid in this calculation, (and I understand not wanting to go back to do the non-parametric bootstrap for all of this), at least mention why it was not included.

The results presented in Table 1 are from the secondary screen, not the mean of the primary and secondary screens. As indicated in the first paragraph of the Results section, the FA values varied from experiment to experiment. Therefore, we cannot add confidence intervals. A positive control (w females of the same genetic background crossed with *da-Gal4, UAS-CycG* ^{ΔP} /*TM6c,Sb* males), as well as a negative control (w females of the same genetic background crossed with *da-Gal4/TM6c,Sb* males) were systematically included in each experimental series to serve as intrinsic standards.

December 20, 2025

RE: GENETICS-2025-308768R1

Dr. Frederique Peronnet
Centre National de la Recherche Scientifique
Development, Adaptation and Aging (DEV2A)
4, place Jussieu
Paris 75005
France

Dear Dr. Peronnet:

Congratulations, your manuscript titled "Genome-wide screen for deficiencies modifying Cyclin G-induced developmental instability in *Drosophila melanogaster*." is accepted for publication in GENETICS! Many thanks for submitting your research to the journal.

In my final reading, I noted a few small edits that should be corrected when submitting the final files:

L50: delete "and" before "the Hippo"

L51: delete comma after CyclinG

L56: change "that" to "which"

L207-208: only 13 increasing deficiencies were identified, so how could 16 be shared with the other study?

L220: change comma after (Figure 2C) to a period.

L263: add a comma after (InR/TOR)

To Proceed to Publication:

1. Format your article according to GENETICS style: <https://academic.oup.com/genetics/pages/author-guidelines>
2. Ensure that you comply with data and community resource citation guidelines: <https://academic.oup.com/genetics/pages/author-guidelines#section-5-9-2>
3. Upload your final files at <https://genetics.msubmit.net>
4. Add oupsupport@scipris.com and genetics.oup@novatechset.com (or the domains @scipris.com and @novatechset.com) to your email program's "safe senders" list. You will be contacted by both at various points during the production process.

Notes:

- Your currently-accepted manuscript (unedited, as submitted, reviewed, and accepted) will be published at GENETICS and deposited into PubMed as an Advance Access article. Notify sourcefiles@thegsajournals.org before signing your license if you do not wish to publish your article via Advance Access.
- We invite you to submit an original color figure related to your paper for consideration as cover art. Please email your submission to the editorial office or upload it with your final files. You can submit a small-sized image for evaluation, and if selected, the final image must be a TIFF file 2513px wide by 3263px high (8.375 by 10.875 inches; resolution of 600ppi). Please avoid graphs and small type.
- After files are sent to Oxford University Press we use SciPris to manage article licensing and payment. If you do not have a SciPris account, you will receive an email from no-reply@scipris.com to sign up to use Oxford University Press' author portal. After logging in, follow the online instructions to sign your license and arrange any payment due.

If you have any questions or encounter any problems while uploading your accepted manuscript files, please email the editorial office at sourcefiles@thegsajournals.org.

Sincerely,

Catherine Peichel
Associate Editor
GENETICS

Approved by:

David Greenstein
Senior Editor
GENETICS